



# Quantitative Chemical Assay of Nanogram-Level PM Using Aerosol Mass Spectrometry: Characterization of Particles Collected from Uncrewed Atmospheric Measurement Platforms

Christopher R. Niedek[1,2], Fan Mei[3], Maria A. Zawadowicz[4], Zihua Zhu[3], Beat Schmid[3], Qi Zhang[1,2*]

[1]Department of Environmental Toxicology, University of California, 1 Shields Ave., Davis, California 95616, United States
[2]Agricultural and Environmental Chemistry Graduate Program, University of California, 1 Shields Ave., Davis, California 95616, United States
[3]Pacific Northwest National Laboratory, Richland, Washington 99352, United States
[4]Environmental and Climate Sciences Department, Brookhaven National Laboratory, Upton, New York 11973, United States

Correspondence to: Qi Zhang (dkwzhang@ucdavis.edu)

**Abstract.** Aerosol generation techniques have expanded the utility of aerosol mass spectrometry (AMS) for offline chemical analysis of airborne particles and droplets. However, standard aerosolization techniques require relatively large liquid volumes (e.g., several milliliters) and high sample masses that limit their utility. Here we report the development and characterization of a micronebulization-AMS (MN-AMS) technique that requires as low as 10 µL of sample and can provide quantification of

nanogram level of organic and inorganic substances via the usage of an isotopically labeled internal standard ($^{34}SO_4^{2-}$). Using standard solutions, the detection limits for this technique were determined at 0.189, 0.751, and 2.19 ng for sulfate, nitrate, and organics, respectively. The analytical recoveries for these species are 104%, 87%, and 94%, respectively. This MN-AMS technique was applied successfully to analyzing filter and impactor samples collected using miniature PM samplers deployable on uncrewed atmospheric measurement platforms, such as uncrewed aerial systems (UAS) and tethered balloon systems

(TBS). Chemical composition of PM samples collected from a UAS field campaign conducted at the DOE Southern Great Plains (SGP) observatory was characterized. The offline MN-AMS data compared well with the in situ PM composition measured by a co-located Aerosol Chemical Speciation Monitor (ACSM). In addition, the MN-AMS and ion chromatography (IC) agreed well for measurements of sulfate and nitrate concentrations in the PM extracts. This study demonstrates the utility of combining MN-AMS with uncrewed measurement platforms to provide quantitative measurements of ambient PM

composition with temporal and spatial resolution.

## 1 Introduction

Aerosols play key roles in human health, air quality, and the climate (Jaffe et al., 2020; Kim et al., 2015; Sommers et al., 2014) and the chemistry of the particles is an important determinant of their hygroscopic, radiative, and toxicological properties (Al-Kindi et al., 2020; Calvo et al., 2013; Contini et al., 2021; von Schneidemesser et al., 2015). Detailed information on aerosol

chemistry and how it varies in the atmosphere is necessary for assessing the effects that ambient particles have on the





environment and public health. For example, while the detrimental health effects of $PM_{2.5}$ (particulate matter with aerodynamic diameter less than or equal to 2.5 μm) as a broad class of pollution have long been recognized (Dockery et al., 1993), recent studies have demonstrated different levels of toxicity among different chemical classes as well as aerosols from different sources, suggesting potentially discrete effects of PM with distinct chemical compositions (e.g. (Contini et al., 2021; D'Evelyn

et al., 2021; Groma et al., 2022; Heal et al., 2012; Plummer et al., 2015; Sun et al., 2017)). A thorough understanding of aerosol composition and chemical processes is also necessary for the development and validation of atmospheric chemical transport models and climate models (Shrivastava et al., 2017). Climate models historically have used physical properties of aerosols (e.g., mass concentration and size distributions) to estimate the radiative effects of particles in the atmosphere, but it is now known that understanding the chemical nature of aerosols is also key to improving model simulations of aerosols' direct and

indirect radiative forcing (Gustafsson and Ramanathan, 2016; Liu et al., 2021b; Lou et al., 2020; Ramanathan et al., 2001; Reddington et al., 2017).

Improved techniques for aerosol chemical measurements are needed to more fully understand the effects of aerosols. Field studies on aerosol chemistry are currently performed through several different avenues, each with their own strengths and weaknesses. Ground-based monitoring approaches can use a suite of instrumentation to obtain highly detailed, continuous

measurements of aerosol physical and chemical properties but are usually restricted to single locations. Piloted aircraft have been utilized in numerous field campaigns around the world and have the advantage of flying to a source of aerosols such as wildfire events (e.g. (Permar et al., 2021; Zhang et al., 2018)) and tracking the evolution of aerosol properties as the plumes disperse (e.g. (Akagi et al., 2012; Kleinman et al., 2020)). However, piloted aircraft are costly to deploy, usually have limited ability to characterize the vertical distribution of aerosols near the surface, and have high speeds that restrict the spatial

resolution of the measurements.

Over the past decade, uncrewed atmospheric measurement platforms (UxS), such as uncrewed aerial systems (UAS) and tethered balloon systems (TBS), have been increasingly used for air quality monitoring (Lambey and Prasad, 2021; Villa et al., 2016) to help fill the gaps left between ground-based and traditional piloted aircraft measurements of atmospheric species (Mei et al., 2022). UAS can be deployed where it would otherwise be too dangerous to fly a piloted aircraft, such as under a

forest canopy (Kobziar et al., 2019), or in particularly remote and challenging locations like Artic areas near newly forming sea ice (de Boer et al., 2018). Additionally, UxS offer an effective way to investigate the vertical stratification of atmospheric components like PM, which is vital for reducing uncertainties regarding aerosol-cloud interactions (de Boer et al., 2018; Creamean et al., 2018; Maahn et al., 2017). In comparison, ground-based measurements of aerosols may not be a reliable method to investigate cloud formation at the time of measurement (Shupe et al., 2013) while limited airborne datasets do not

offer the spatial or temporal resolution necessary for a more full understanding of the relationship between cloud properties and aerosols.

*In situ*, high time resolution, and low detection limit measurements of PM chemistry are difficult to achieve with UxS due to the high energy-consumption and weight of the required instrumentation (Brady et al., 2016; Glaser et al., 2003; Hemingway et al., 2017). The studies that demonstrate such measurements are generally limited to particle number and size distributions



(e.g. (Aurell et al., 2021; Bates et al., 2013; Brady et al., 2016; Corrigan et al., 2008; Girdwood et al., 2020; Kezoudi et al., 2021; Villa et al., 2016)). For detailed chemical information, offline analysis of UxS-collected PM is more plausible as this removes the need to have the heavy instrumentation aboard a UxS. However, due to payload restrictions, the samplers on-board uncrewed platforms usually have a low volumetric flow rates that severely limit the total collectible PM mass from UxS (Villa et al., 2016). For example, the US Department of Energy's TigerShark, which can afford several hours of continuous

flight time with a payload of ~100 lbs (Mei et al., 2022), uses a filter sampling system that operates at a flow rate of 2.5 L/min, which is ~400 times lower compared to the high volume air samplers often used for ground-based sampling.

Aerosol mass spectrometry (AMS) is a widely used technique for quantitative measurement of non-refractory (NR) aerosol species such as sulfate, nitrate, ammonium, chloride, and organics (Canagaratna et al., 2007; DeCarlo et al., 2006). While the application of AMS has primarily been real-time measurements (e.g. (Fountoukis et al., 2014; Li et al., 2017; Zhou et al.,

2020)), in recent years, an increasing number of studies have reported the usage of AMS for offline analysis of PM samples to describe long-term chemical characteristics of PM or to examine the sources and chemical properties of water-soluble and insoluble components (e.g. (Bozzetti et al., 2017; Daellenbach et al., 2016; Ge et al., 2017; Li et al., 2021, 2020; Moschos et al., 2018; Sun et al., 2011; Vlachou et al., 2018)).

The AMS is a highly sensitive instrument with 1-min detection limits of ~20 ng m$^{-3}$ for organics and as low as 2.9 ng m$^{-3}$ for

nitrate at an air sampling flow rate of ~ 0.1 L min$^{-1}$ (DeCarlo et al., 2006). However, the amount of PM mass that needs to be collected for offline AMS analysis is dependent on the liquid volume and concentration required for stable particle generation in the size range needed for AMS sampling. Since the nebulization efficiency (i.e. the ratio between the mass detected by the AMS compared to the mass of solute nebulized) of the common aerosol generation systems is low, e.g., ~ 0.02% for an ultrasonic atomizer utilized by O'Brien et al. (O'Brien et al., 2019), liquid volumes of several milliliter and tens of micrograms

of sample mass are usually required for continuous aerosol generation and AMS analysis (O'Brien et al., 2019; Sun et al., 2011). Given a typical ambient PM concentration of 10 µg m$^{-3}$, several cubic meters of air need to be sampled to meet this mass requirement, which is very difficult to achieve with many UxS. Taking the characteristics of the TigerShark as an example, the on-board PM filter sampler has a flow rate of 2.5 L/min (Mei et al., 2022), thus requires 400 min of flight time to sample 1 m$^3$ of air. This flight time is not practical to many UAS currently used and as such it is necessary to substantially

increasing aerosol generation efficiency for the AMS analysis of UAS collected samples.

In this study, we develop a novel analytical technique that combines isotopically-labeled internal standardization, micronebulization, and high-resolution aerosol mass spectrometry to achieve quantitative analysis of nanogram-level of PM in liquid samples. This MN-AMS technique expands the utility of offline AMS analyses by dramatically reducing the required liquid volumes needed for stable aerosol generation. In addition, this method uses sulfur-34 labeled ammonium sulfate (A$^{34}$S)

as the internal standard to achieve quantification of liquid concentration based on AMS measurements. While ammonium sulfate has frequently been used as an internal standard in lab studies (e.g. (Jiang et al., 2021; Ma et al., 2021; Yu et al., 2014)), it cannot be used for ambient samples without an independent measurement of sulfate concentration. We also present the



application of this analytical method to ambient PM samples collected using UAS instrumentation, including from a recent UAS field campaign.

## 2 Materials and Methods

### 2.1 Chemicals

Chemicals were used as received. Sucrose (ACS grade), sulfuric acid (ACS Plus grade), and methanol (LC-MS grade) were from Fisher Scientific. Ammonium sulfate (AS; ACS reagent grade) and A$^{34}$S (>98 % $^{34}$S) were from Millipore Sigma. Anhydrous sodium carbonate was from Alfa Aesar. All chemical solutions were prepared using ultrapure water (Milli-Q water; ≥ 18.2 MΩ cm).

### 2.2 UAS sample collection and site description

Ambient aerosol samples examined in this study were collected at two locations: the Pacific Northwest National Lab (PNNL) in Richland, Washington and the central facility of the Southern Great Plains (SGP) observatory, which is operated by the Department of Energy's (DOE) Atmospheric Radiation Measurement (ARM) program, and located near Lamont, in north-central Oklahoma. PM$_{2.5}$ samples were collected on two types of substrates: PTFE filters and aluminum impactor stubs. The PTFE filters were installed with a time-resolved filter sampler (Model 9401, Brechtel) designed for deployment on UxS (Mei and Goldberger, 2020). The aluminum impactor stubs were installed inside a custom-built growth tube to collect droplets generated from a moderated aerosol growth inside a water-based Condensation Particle Counter (CPC) (Hering et al., 2014). The sampling rates are 2.5 and 0.3 L min$^{-1}$ for filter and impactor collection, respectively. Both substrates were precleaned using a methanol (HPLC grade, Fisher Scientific) wash followed by ultrasonication in purified water for 10 min. All samples were acquired on the ground except for the filter sample from the SGP site which was sampled across multiple UAS flights for 15 hours during November 2021. Details of the SGP UAS campaign are described in Mei 2022 (Mei et al., 2022). A table of sampling information can be found in the Supplemental Information (Table S1). In addition, lab generated aerosols composed of sucrose and ammonium sulfate were collected using the same samplers for initial method development.

On November 16 and 17, 2021, additional aerosol samples were collected on multiple silicon substrates using a four-stage impactor (Sioutas Personal Cascade Impactor, SKC Inc.), operating at a flow rate of 9 L min$^{-1}$. The impactors were deployed on the roof of the Aerosol Observation System trailer (AOS, 10 m above the ground) for two hours per day. Note that the aerosol particles collected by the impactor were analyzed offline by a time-of-flight secondary ion mass spectrometry (ToF-SIMS) (detailed in 2.3.4).



## 2.3 Chemical analyses of PM samples

### 2.3.1 Extraction of PM

A schematic overview of the PM sample extraction and analysis steps can be found in Figure 1a. For the extraction of a PTFE filter sample (13 mm in diameter), a portion of the filter (punched out with a 5/32-inch diameter puncher) is placed in a microcentrifuge tube (1.5 ml conical) along with 100 µL methanol (LC-MS grade) and subjected to ultrasonication in an ice bath for 15 minutes. Methanol was chosen as the extraction solvent to increase the proportion of organic material that could be removed from the filters. After this first sonication, 300 µL of 1 mg $L^{-1}$ $^{34}SO_4^{2-}$ internal standard solution was added, the combined solution was sonicated for 30 minutes, and then diluted to 1 mL with 1 mg $L^{-1}$ $^{34}SO_4^{2-}$. To extract the impactor stub samples, where particles were collected on the surface in a ~ 2 mm diameter spot, 15 µL of methanol was added to the impactor surface and the surface was gently scraped and the resulting solution was transferred into a microcentrifuge tube. This procedure was repeated 3 times to ensure quantitative transfer of PM into the microcentrifuge tube and the combined solution was sonicated for 15 minutes. Then, 1 mg $L^{-1}$ $^{34}SO_4^{2-}$ was added to bring the final volume to 500 µL and sonicated again for 15 minutes. All ultrasonication procedures were performed at 0 ºC to prevent heat-induced degradation. Finally, the filter and impactor extracts were filtered using syringe filters (0.45 µm PTFE) to remove insoluble materials and stored frozen at -20ºC prior to chemical analysis.



(a)

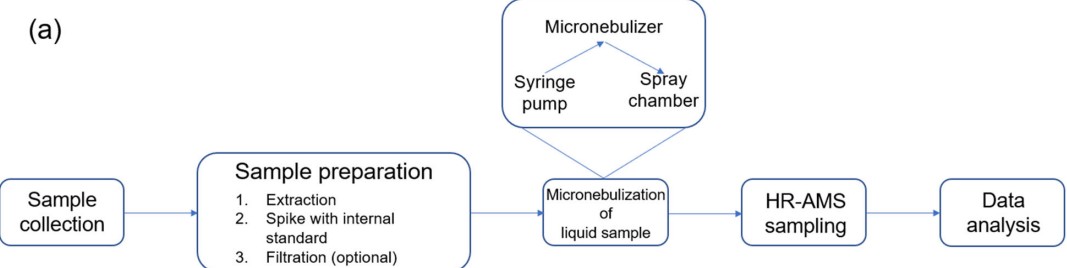

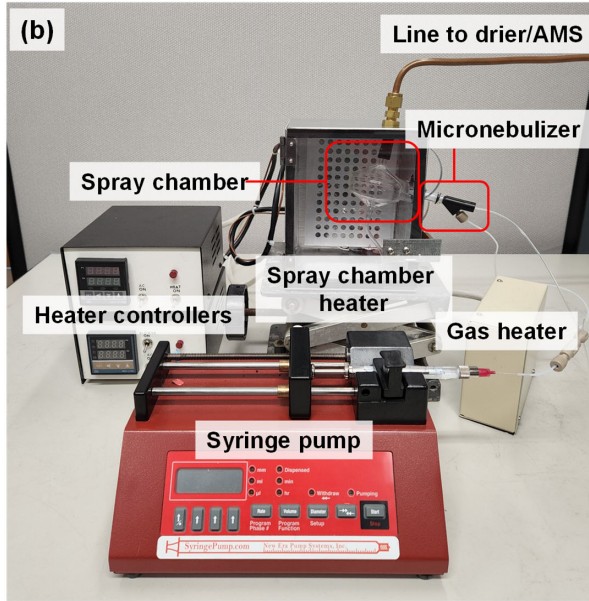

**Figure 1. a) A schematic overview of our microextraction and analysis methodology. b) A picture of the micro-flow nebulization setup.**

### 2.3.2 Aerosol mass spectrometry (AMS) analysis

A high-resolution time-of-flight AMS (HR-AMS; Aerodyne Res. Inc.) was used to characterize the bulk chemical composition of the filter and impactor extracts. The HR-AMS was typically operated in "V" mode (with a mass resolution of ($m/\Delta m$) of ~3000) with 1 min averaging. When the fast sampling mode was used, the averaging time was decreased to 1 sec. Prior to sampling with the AMS, liquid extracts were aerosolized using a micronebulization assembly, which is pictured in Figure 1b and discussed in detail in section 3.1.


### 2.3.3 Ion chromatography (IC) analysis

An ion chromatograph (Metrohm 881 Compact IC Pro) with a conductivity detector was used for measurement of anions. The anion IC was equipped with a Metrohm A Supp 7 250/4.0 column, 3.6 mM $Na_2CO_3$ was used as the eluent, and 0.1 M $H_2SO_4$ was used as the suppressor solution. Calibration curves of $SO_4^{2-}$ and $^{34}SO_4^{2-}$ were created with a concentration range of 0-500 $\mu g\ L^{-1}$ in terms of $SO_4^{2-}$ or $^{34}SO_4^{2-}$. $SO_4^{2-}$ and $^{34}SO_4^{2-}$ co-elute (Figure 2a) so separate calibration curves are necessary to quantify $SO_4^{2-}$ and $^{34}SO_4^{2-}$ in samples containing both species.

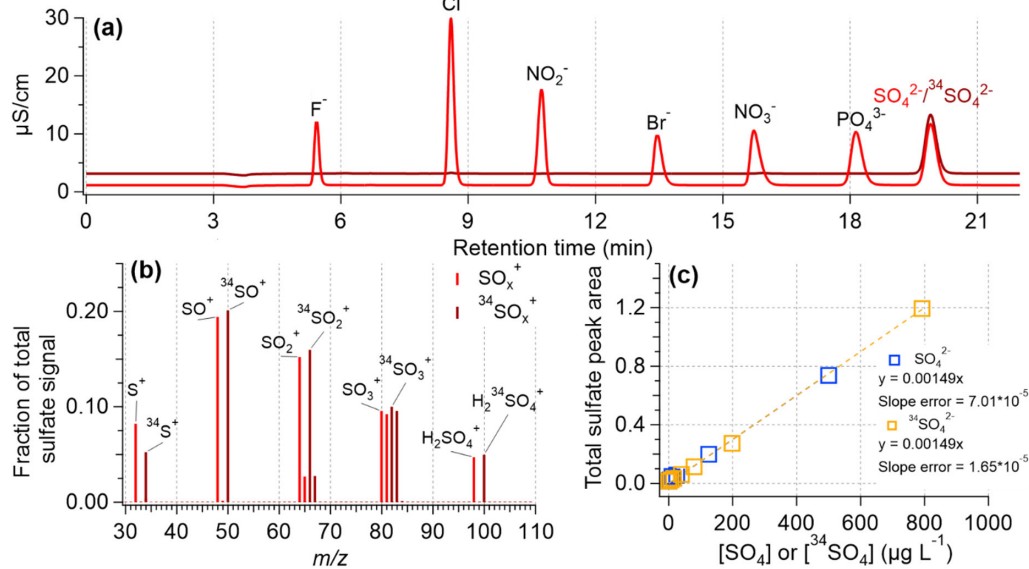


**Figure 2. Assessment of the instrumental response to $SO_4$ and $^{34}SO_4$. a) IC chromatograms of an anion standard mix (in light red) and a $^{34}SO_4^{2-}$ standard. The $SO_4^{2-}$ and $^{34}SO_4^{2-}$ concentrations were 3 mg $L^{-1}$ in each solution. b) HR-AMS mass spectra for a solution containing equal concentrations of $SO_4$ and $^{34}SO_4$. The HR-AMS signal for the $^{34}SO_4$ ions follow the expected trends based on the standard $SO_4$ ions. c) IC standard calibration curves of sulfate ($SO_4^{2-}$) and isotopically labeled sulfate ($^{34}SO_4^{2-}$).**


### 2.3.4 Secondary ion mass spectrometry (SIMS) analysis

SIMS measurement of PM composition was performed at the Environmental Molecular Sciences Laboratory (EMSL), which is located at PNNL. A TOF.SIMS5 instrument (IONTOF GmbH, Münster, Germany) was used. A 25 keV pulsed $Bi_3^+$ beam was used as the analysis beam to collect SIMS spectra. The $Bi_3^+$ beam was focused to be ~0.4 μm diameter and scanned over

a $100 \times 100$ μm$^2$ area on the aerosol particles collected on silicon wafers using the delay extraction mode. The mass resolution (m/Δm) of the SIMS was in a range of 3000-5000. Data reconstruction was conducted using the SurfaceLab 6 software (Version 6.3, IONTOF GmbH, Germany). Region-of-interest (ROI) reconstruction was performed, in which only signals from





aerosol particles were reconstructed as new spectra, while the signals from the silicon substrate were excluded. Mass calibration was carried out using characteristic peaks, e.g., $CH_3^+$ (m/z 15), $C_2H_3^+$ (m/z 27), $C_3H_3^+$ (m/z 39), and $Bi^+$ (m/z 209) in positive

ion spectra, and $CN^-$ (m/z 26), $C_3^-$ (m/z 36), $NO_3^-$ (m/z 62), and $SO_4H^-$ (m/z 97) in negative ion spectra.

**2.4 Data processing**

The HR-AMS data were processed using the standard AMS data analysis toolkits (SQUIRREL v1.63H and PIKA v1.23H). Although a high-capacity silica gel drier was used, a large particle water signal was measured. To avoid potentially overestimating the organic water signal, the $H_xO^+$ signals were parameterized using the standard method for HR-AMS ambient

data processing: $H_2O^+ = 0.225 \times CO_2^+$, $HO^+ = 0.25 \times H_2O^+$, and $O^+ = 0.04 \times H_2O^+$ (Aiken et al., 2008). In addition, since purified nitrogen was used as the carrier gas in this study, the $CO^+$ signal was also parameterized using the Aiken method for ambient aerosol: $CO^+ = CO_2^+$ (Aiken et al., 2008).

In order to separately quantify $SO_4$ and $^{34}SO_4$ by the HR-AMS, several modifications were made to the PIKA analysis procedures. First, all sulfate-related ions, e.g., $H_{y\geq0}{}^{34}SO_{x\geq0}{}^+$ (Allan et al., 2004) with the $^{34}S$ isotope were unconstrained so the

signals would not be parameterized based on the parent isotope and natural isotopic abundances. A high-resolution fragmentation wave was also created to represent $^{34}SO_4$. This pattern was similar to the standard fragmentation pattern for sulfate except that the sulfate-associated $H_2O^+$ signal was parameterized to the $^{34}SO_2^+$ and $^{34}SO^+$ ions and the parameterizations for the S and $^{33}S$ signals were removed. A table containing the fragmentation pattern can be found in the supplemental information (Table S2). Last, a new ion family containing all sulfate-relevant $^{34}S$ ions was created, separate from the standard

ion family containing all of the sulfate-relevant ions.

In order to determine the spiked $^{34}SO_4$ by the AMS, the natural abundance of $^{34}S$ present in the native sulfate must be subtracted out of the measured $^{34}SO_4$. This adjusted $^{34}SO_4$ concentration ($[^{34}SO_4]_{AMS, adj}$) is calculated using the natural isotopic abundance of $^{34}S$ (= $0.0447 \times {}^{32}S$):

$$[^{34}SO_4]_{AMS,adj} = [^{34}SO_4]_{AMS} - [SO_4]_{AMS} \times 0.0447 \qquad (1)$$

where $[^{34}SO_4]_{AMS}$ and $[SO_4]_{AMS}$ are the HR-AMS measured aerosolized concentrations ($\mu g\ m^{-3}$) of $^{34}SO_4$ and $SO_4$, respectively. The liquid concentration of component X ($[X]_{liquid}$; $\mu g\ L^{-1}$), e.g., organics, nitrate, chloride, and the native $SO_4$, is calculated as:

$$[X]_{liquid} = [X]_{AMS} \times \left(\frac{[X]}{[^{34}SO_4]_{AMS,adj}}\right) \qquad (2)$$

where $[^{34}SO_4]$ is the known concentration ($\mu g\ L^{-1}$) of the internal standard ($^{34}SO_4{}^{2-}$) in the liquid sample.

Then, the ambient concentration ($\mu g\ m^{-3}$) of the sampled PM components ($[X]_{ambient}$) can be calculated:

$$[X]_{ambient} = \frac{[X]_{liquid} \times V_{extract}}{V_{air} \times 1000} \qquad (3)$$

where $V_{extract}$ is the total extract volume (ml), $V_{air}$ is the total volume of air sampled ($m^3$), and 1000 is a unit conversion factor.



IC analysis was performed in this study as an independent check for the accuracy of AMS quantification. For IC analysis, calibration curves of $SO_4^{2-}$ and $^{34}SO_4^{2-}$ were generated for a sulfate concentration range of 0-500 µg L$^{-1}$. Calibration curves

fitting parameters were used for later separation of the $SO_4^{2-}$ and $^{34}SO_4^{2-}$ signal in samples containing both ions. For calculation of $SO_4^{2-}$ in ambient samples that have been spiked with $^{34}SO_4^{2-}$, the IC peak area is first used to estimate the $SO_4^{2-}$ liquid concentration ($[SO_4]_{est}$) assuming the signal is solely from $SO_4^{2-}$:

$$[SO_4]_{est} = \frac{(SO_{4PA} - b)}{m} \tag{4}$$

where $SO_{4,PA}$ is the IC-measured peak area, and m and b are the linear regression slope and intercept, respectively, from the

calibration curve of $SO_4^{2-}$. The contribution from the $^{34}SO_4^{2-}$ internal standard is then subtracted out to determine the true liquid concentration of $SO_4^{2-}$.

### 3 Results and discussion

### 3.1. Assessment of the micronebulization-AMS Technique

#### 3.1.1 Micro-flow nebulization system and interfacing with AMS

Due to the low PM mass that can be collected by many weight-limited aerial platforms, micronebulization techniques that can achieve ultra-low flow rates, thus requiring significantly lower sample masses compared to common, collision-type atomizers, are sorely needed for offline AMS analysis of such samples. O'Brien et al. reported the utilization of a micronebulization system based on ultrasonic atomization to enable offline AMS analysis of microgram-level samples for atmospheric research (O'Brien et al., 2019). However, this method suffers severe sample loss upstream of the AMS and the nebulization efficiency

(NE), which is defined as the ratio of mass measured by the AMS to the known mass of nebulized analyte, was found to be only 0.02-0.06% (O'Brien et al., 2019). Apparently, the sensitivity of offline AMS methods can be substantially increased through improving the efficiency of the aerosol generation interface.

Figure 1b shows a picture of the micro-flow nebulization system developed in this study that can be interfaced directly with the AMS to allow sensitive detection and chemical characterization of nanogram-level samples. This system consists of a

syringe pump that delivers liquid to a concentric nebulizer made of Teflon at a predefined flow rate (e.g., 100 µl/min). The liquid is nebulized using pressurized gas such as high purity nitrogen or argon (50 psi) and the resulting fine mist enters a glass cyclonic spray chamber, where large droplets are removed. Both the gas line and the spray chamber are mildly heated to facilitate droplet evaporation and minimize condensation on the spray chamber wall. The resulting aerosol then passes through a silicate diffusion drier before entering the HR-AMS.

The MN-AMS method was initially evaluated using standard solutions composed of sucrose, AS, and A$^{34}$S with additional method validation using IC analysis. Figure S1 compares the AMS mass spectra of a solution atomized using a standard, collision-type atomizer (TSI 3076) and the micronebulizer. The organic and inorganic mass spectra derived from each





atomization system show a high degree of similarly ($r^2 = 0.99$), indicating that the micronebulization system does not introduce any artifacts compared to a standard atomizer.

Since the transmission efficiency of the AMS aerodynamic lens is size-dependent and is nearly 100% for particles in the diameter range of ~ 100 – 500 nm (Jayne et al., 2000; Liu et al., 2007), it is necessary to control the aerosol sizes generated from the micronebulization system to maximize the overall sensitivity of the MN-AMS system. Factors affecting the size distribution of the droplets from the nebulizer, thus the dry particle sizes, include the total solute concentration and liquid sample flow rate. Figure S2 shows that, at total solute concentrations of 1- 7 mg L$^{-1}$, the mode vacuum aerodynamic diameter

($D_{va}$) (DeCarlo et al., 2004) of the generated particles is ~100-200 nm, well within the 100% transmittance range of the AMS (Liu et al., 2007). Decreasing the solute concentration to a low value less than 1 mg L$^{-1}$ may cause significant reduction of MN-AMS sensitivity as the measured particle size may have a considerable fraction of particle mass outside of the 100% transmission range of the AMS ($D_{va}$ ~ 100 – 500 nm). Lowering the liquid flow rate can decrease the mode size distribution as well (see Fig. S2b, S2c), but when operated under the flow rate specified for the nebulizer (i.e. 50 µl min$^{-1}$), the effect is

small compared to lowering the total solute concentration.

**3.1.2 Nebulization efficiency**

The nebulization efficiency (NE) of the MN-AMS was determined by nebulizing 400 µL of solution with varying concentrations of sucrose, SO$_4$, and $^{34}$SO$_4$ (where the total solute concentration was kept constant at 9 mg L$^{-1}$) and integrating the AMS-measured mass of the individual components over the entire length of sample nebulization (O'Brien et al., 2019).

Figure 3a shows the variation in NE for organics and SO$_4$ as a function of the nebulized mass which was in turn varied by dilution with $^{34}$SO$_4$ solution (to keep the total solute concentration constant) as well as by decreasing the syringe pump flow rate. A NE of 0.93 ~ 1.2 % was determined for the NM-AMS system. One of the factors responsible for the low NE is that the concentric nebulizer requires around 50 psi of gas pressure to function properly. The high pressure meant that the aerosol flow rate out of the nebulizer is notably higher than the AMS inlet flow rate, meaning the AMS is subsampling the total aerosol

mass. Another factor is loss of nebulized mass due to condensation inside the spray chamber, and this was partially corrected by mildly heating the spray chamber and the gas line (see Fig. 1a). Loss of aerosols inside the diffusion dryer may also be a factor as well. Depending on the design of the experiment, these different factors may not be tunable, whereas syringe pump flow rate and solution composition can be controlled to maximize the NE of the MN-AMS system.

In addition to optimizing the efficiency of aerosol generation, the AMS sampling frequency is also a critical factor determining

the minimum sample volume, thus the MN-AMS detection limits. With very low sample volumes, the Fast-MS mode of the AMS is particularly useful as it reduces the sampling time to 1 second or less (Kimmel et al., 2011), thus requiring much less aerosol mass compared to the standard sampling mode which acquires mass spectral data over at least 6 seconds (i.e., 3 seconds each on the chopper-open and the chopper-closed positions) (DeCarlo et al., 2006). As shown in Figure S3, for MN-AMS setup reported here, the Fast-MS mode provides highly reproducible measurements of different chemical components in the



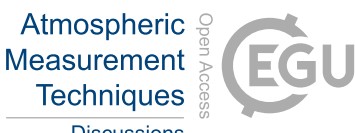

liquid sample and the liquid concentration of organics measured using both modes are comparable when normalized to the

known concentration of $^{34}SO_4$.

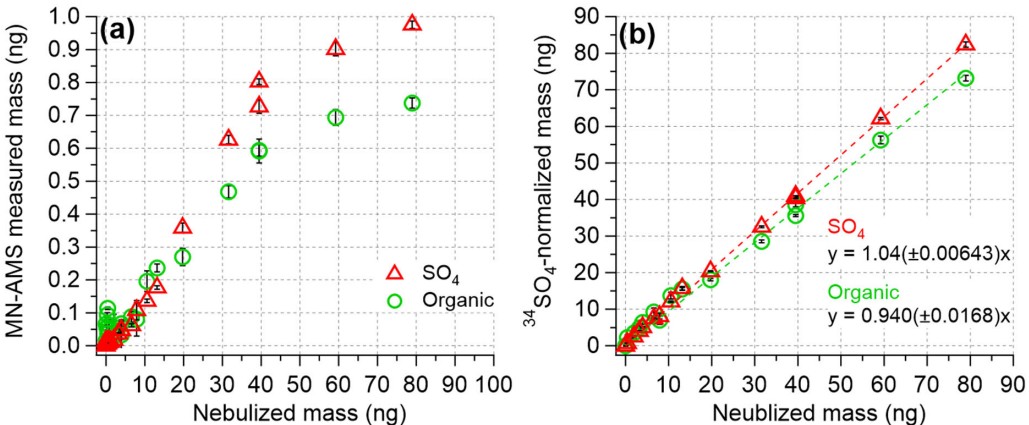

**Figure 3. a) The MN-AMS measured mass of organics and SO₄ compared to the expected mass of sucrose and SO₄. The MN-AMS measured mass is not linearly proportional to the nebulized mass, but this can be corrected with internal standardization (as in Fig. 1b). The ratio between the two values gives the nebulization efficiency of each component. b) The ³⁴SO₄-normalized mass of organics and SO₄ compared to the nebulized mass.**

### 3.1.3 Quantification via internal standardization

Due to variations in factors affecting the AMS signal intensities, such as nebulization efficiency discussed above, the use of an internal standard (IS) is necessary for quantitative analysis of liquid samples using AMS (Jiang et al., 2021; O'Brien et al.,

2019; Yu et al., 2016). An IS is a chemical substance that is added in a constant amount to every sample analyzed, including the samples, the blank, and calibration standards. It allows for quantification of other measured species and for correcting variabilities in NE and AMS detection sensitivity. Isotopically labeled internal standards are commonly used for mass spectrometry as they are very similar to the chemical species of interest in the samples and the effects of sample preparation should, relative to the amount of each species, be the same for the signal from the internal standard as for the signal from

analyte. The use of isotopically labeled internal standards for AMS analysis, specifically $^{15}NO_3$, has been explored previously (O'Brien et al., 2019). In the present study, we chose $^{34}SO_4$ as the IS due to the well-characterized fragmentation behavior of sulfate in the AMS, fragment ions that are separated well from isobaric ions, and compatibility in both the HR-AMS and IC systems. Furthermore, the low volatility of ammonium sulfate prevents positive artifacts due to the evaporation loss of the internal standard.

A key use of isotopically labeled sulfate is the quantification of ambient PM components, particularly sulfate, in the HR-AMS. Figure 2a shows that $^{34}SO_4^{2-}$ behaves the same as the natural sulfate ($SO_4^{2-}$) in the AMS, producing nearly identical fragmentation patterns. Additionally, while $SO_4^{2-}$ and $^{34}SO_4^{2-}$ co-elute in IC (Figure 2b), the response factor for each form of sulfate is nearly identical (Figure 2c). The effectiveness of an internal standard for AMS quantification of liquid samples is



demonstrated in Figure 3, where the $^{34}SO_4$-normalized masses of organics and $SO_4$ accurately reproduce the known, nebulized

masses of organics and sulfate (Fig. 3b) while the unnormalized measurement data do so poorly (Fig. 3a).

**3.2. Evaluation of the MN-AMS method using PM samples collected by UxS samplers**

In order to evaluate the MN-AMS method for UxS sample analysis, a total of eight filters and three impactor samples of ambient PM were collected from PNNL and analyzed. The filters were sampled for 3 h periods, corresponding to 0.45 $m^3$-air, and the impactors were sampled for variable lengths of time corresponding to 0.036 – 0.34 $m^3$-air (Table S1). Blank filters and

impactor stubs were also processed and analyzed in the same manner as the sampled filters and impactors. Since low volatility organic matter in organic solvents may produce background signals in the AMS, we used ultrapure, LC-MS grade methanol for PM extraction. Testing with varying concentrations of methanol in 1 ppm $^{34}SO_4^{2-}$ solution revealed that LC-MS grade methanol still generates organic background signals in the AMS analysis, but final methanol concentrations at or below 10 % gave a consistent and acceptably low background. The $^{34}SO_4$-normalized liquid concentration of organics in the blanks were

subtracted from the filter and impactor data. The organic contribution from the blank filter and impactor were at most 15 % and 30 % of the organic signal from this set of ambient filters and impactors, respectively.

As the last filter (PNNL_F8) and impactor (PNNL_I3) were sampled during the same time period (Table S1), the chemical composition of each can be compared as a means of assessing biases in the sampling system or in our extraction procedure (Figure S4). Overall, the chemical compositions of the filter extract and the impactor extract are similar (Fig. S4a-

c), although the HR-AMS spectrum of impactor extract shows relatively enhanced $C_xH_y^+$ ions and several higher *m/z* $C_xH_yO_{>1}^+$ ions compared to the MS of the filter extract. Given that these ions are also significant in the methanol solvent blank mass spectrum, a potential contamination from methanol was possibly the reason. However, it is also important to note the differences in extraction technique used for the filter and impactor samples as described in section 2.2. For example, a significant difference in the volume of air sampled for the filter vs. the impactor over a given time period led to much less

collected PM on the impactor stub. In addition, while both sample types were initially extracted with pure methanol, the final methanol concentration is higher in the filter extracts compared to the impactor extracts (6.7 % vs. 2.2 %), leading to potentially different contributions from methanol residuals. Despite these potential confounding issues, our results indicate that the filter and impactor samples are chemically quite similar.

**3.3. Chemical characterization of aerosol Samples Collected from a UAS Campaign and intercomparison with**
**collocated measurements**

A field campaign was conducted at the SGP site to examine techniques of UxS measurements and data analysis (Mei et al., 2022). Figure 4 is a UAS flight track (colored with the aerosol total number concentration from a CPC) overlapped with the United States Geological Survey (USGS) national map of the SGP site and surrounding area. The Central Facility is located in a rural environment with several large urban areas located within 200 km. A refinery and a coal-fired power plant are located

within 50 km of the central facility (Liu et al., 2021a; Sisterson et al., 2016). The diversity of land use at the SGP site causes



a diversity of air masses originating from anthropogenic, biogenic, and biomass burning sources (Liu et al., 2021a; Parworth et al., 2015).

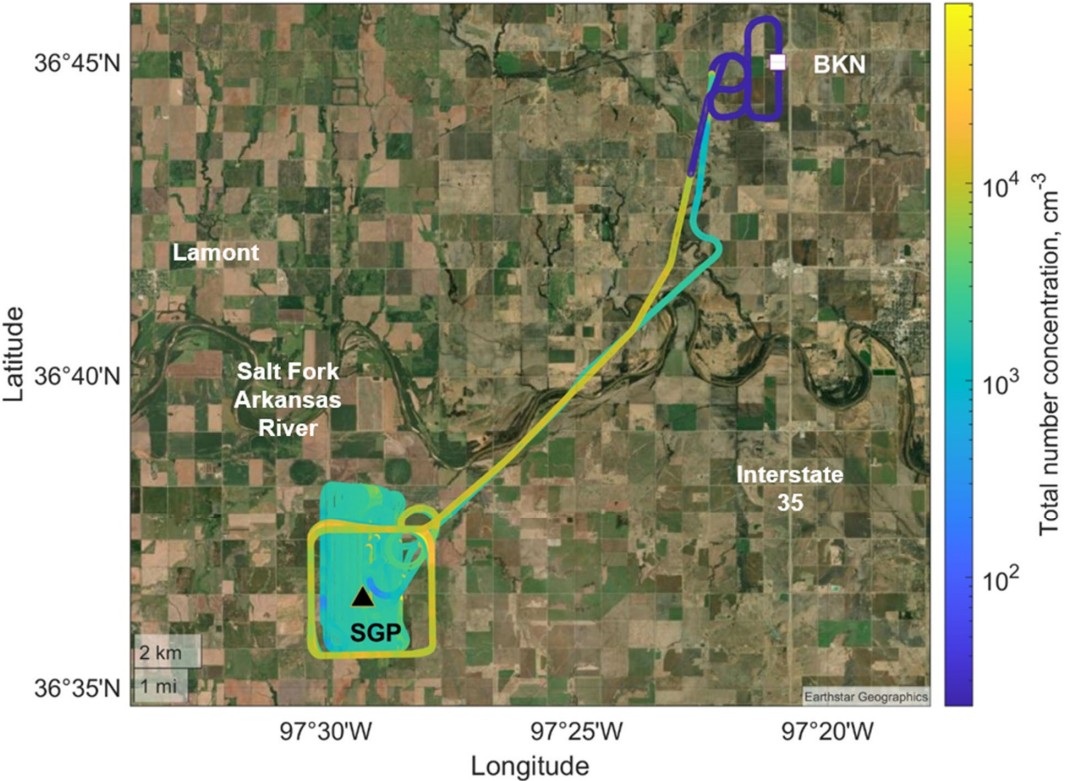

**Figure 4. A typical flight track of one UAS flight (recorded on 2021/11/13) overlapped with the USGS national map. The UAS took off from Blackwell Municipal airport (BKN; white square) and sampled near the Central Facility (black triangle) of the SGP observatory. The flight track is colored with the total particle number concentration from an onboard CPC. Other sampling days used a partial version of this flight pattern due to weather limitations. Ground sampling occurred at the Central Facility.**

Ground and air samples were collected at SGP and analyzed using the MN-AMS technique. Additionally, an Aerodyne quadrupole aerosol chemical speciation monitor (ACSM) provided continuous measurements of non-refractory submicrometer aerosols (NR-PM$_1$) at the SGP site during the UAS deployment period. While there are differences in instrumentation and data analysis techniques used between the MN-AMS and the ACSM, the particle vaporization and ionization mechanisms are the same, so comparisons can still be reasonably made. Figure 5a shows the time series of ACSM-measured NR-PM$_1$ species, along with the corresponding sampling periods for the impactors and UAS filter samples. The MN-AMS data and the ACSM data show similar NR-PM composition during the time periods the UAS filter and impactor samples were collected (see inset



pie charts in Fig. 5a). The organic mass spectra patterns from the two approaches are similar too for the indicated sampling
periods (Fig. 5 b-e and Figure S5 a-d).





**Figure 5. Comparison of offline PM$_{2.5}$ composition measured offline using the MN-AMS vs. the real-time PM$_1$ measured using a Quadruple ACSM. a) Time series of the ACSM-measured NR-PM1 species at the SGP site. The inset pie charts compare the**
**composition measurements by the MN-AMS (darker colors) and the ACSM (lighter colors) for the same time periods. b-e) High mass resolution organic spectra colored by 8 ion categories for the SGP filter and impactors measured by MN-AMS. The average atomic ratios in the organics of each samples are shown in the legends.**

The MN-AMS data can be used to back-calculate the ambient PM mass concentration (see section 2.4 for details). The ACSM
data suggested a reasonably neutralized particle mass during the sampling period, and this information was used to correct the ammonium concentration in the offline samples (due to the addition of isotopically labeled ammonium sulfate) by assuming the measured SO$_4^{2-}$ and NO$_3^-$ was in full charge balance with the NH$_4^+$. The comparison between the MN-AMS estimated ambient PM loading and the ACSM measured loading can be found in Figure 6. The PM loadings are within 20% except for impactor 1 where the MN-AMS measurement reports a notably higher organic mass loading than that measured by the ACSM.
Besides the chemical differences between PM$_{2.5}$ and PM$_1$, the discrepancy could also be due to contamination which could occur during sample collection and processing. Nevertheless, the general trend in the total PM loading is quite similar between the two instruments, suggesting the MN-AMS technique is recapturing the real-time measurements to a large degree.

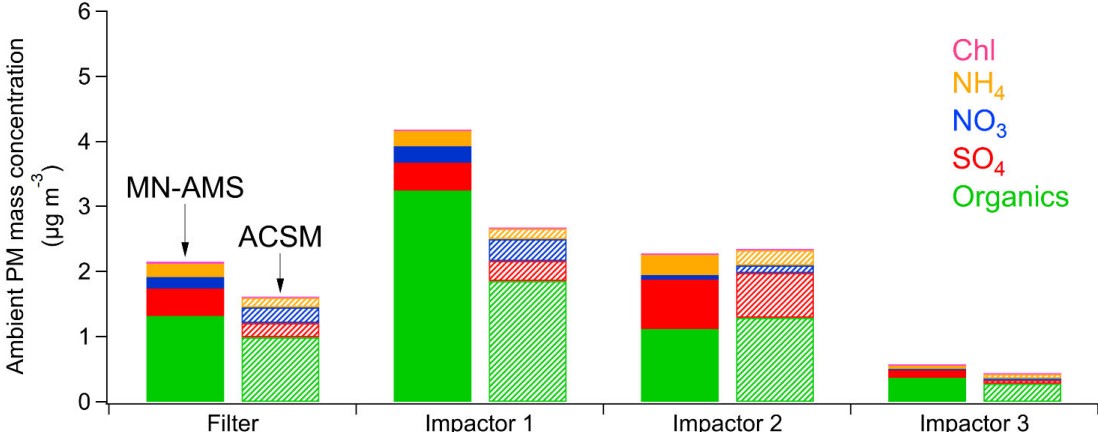

**Figure 6. Comparison of the ACSM-measured ambient NR-PM$_1$ mass concentration and the MN-AMS back-calculated ambient**
**NR-PM$_{2.5}$ mass concentration at the SGP site. Sampling time periods for the filter and impactors are shown as shaded areas in Fig. 5a.**

In addition, the high mass resolution of the MN-AMS allows richer chemical information of PM to be obtained compared to the ACSM. For example, the detection of nitrogen-containing organic species in the filter and impactor samples is noteworthy as organonitrates have been previously suspected to be present at SGP (Parworth et al., 2015). As a fraction of total measured
organic ions, the SGP PM$_{2.5}$ samples contained 3.8 to 8.5 % C$_x$H$_y$N$_{1,2}^+$ and C$_x$H$_y$NO$^+$ type of ions (Fig. 5 b-e and Figure S6a-d). The filter sample (collected aboard a UAS) contained primarily C$_x$H$_y$N$^+$ ions with C$_2$H$_4$N$^+$ being the highest signal in the





family. The impactor samples had more $C_xH_yNO^+$ ions with $C_2H_5NO^+$ as the most dominate in the family. These results highlight the importance of N-containing organic aerosols in the SGP region. This finding is confirmed by SIMS analysis of the same PM samples. An example of one of the SIMS unit mass spectra collected is shown in Figure S7a with a selection of

high-resolution ion fittings shown in Figure S7b. The large differences in sampling and ionization mechanisms between the two instruments precludes specific chemical comparisons, but the general characteristics of the data can be compared. The SIMS detected a large number of nitrogen-containing organics as well as many $CH^+$ ions and few $CHO^+$ ions. Most of the detected nitrogen species produce $C_xH_yN^+$ type of ions with a small number of $C_xH_yNO^+$. The detection of nitrogen-containing organics using both the MN-AMS technique and SIMS helps to validate the use of the MN-AMS technique and, alongside a

previous publication from Parworth et al. (Parworth et al., 2015), suggests further study of the nitrogen-containing organics at the SGP site is warranted.

### 4. Conclusions

This study evaluated a novel MN-AMS technique and demonstrated its utility for the quantitative, chemical analysis of low mass, low volume PM samples such as those collected from UxS platforms. The micronebulization technique can continuously

generate aerosols from tens of microliter sample volumes, a large improvement from commonly used atomization systems that require volumes in the range of 5-15 mL. Nebulization efficiencies, detection limits, and recoveries (calculated using HR-AMS data) for sulfate, nitrate, and organics are summarized in Table S3. Nebulization efficiencies are in the range of 0.93-1.2 % (depending on solute concentration and syringe pump flow rate), an order of magnitude higher than the nebulization efficiencies reported by O'Brien et al. using an ultrasonic atomization system (O'Brien et al., 2019). A main cause for the low

NE in MN-AMS is analyte losses due to condensation of droplets inside the spray chamber. In addition, particle loss inside the drier, subsampling of the total atomization output by the AMS, and partial AMS lens transmission of particles outside of the optimal transmission size range ($D_{va}$ of $\sim 100 - 500$ nm (Liu et al., 2007)) may contribute to the low NE as well. The detection limits of this MN-AMS method for organics, nitrate, and sulfate are on the order of 1 ng, with analytical recoveries ranging between 87 - 104 %.

A key advantage of the MN-AMS technique is the requirement of lower liquid volume (as low as 100 µL) for stable aerosol generation, which translates to substantially lowering the required initial PM mass that must be collected for offline AMS analysis. As a result, this technique meets the needs from the growing desire for atmospheric UxS sampling and the widespread use of the AMS for offline chemical analysis of PM samples. As a proof-of-concept, a small number of UAS-collected PM samples were analyzed using the MN-AMS technique. Further analysis of UAS samples is advisable to explore the utility of

the MN-AMS technique for investigating the ambient aerosol chemical distribution with improving sampling time resolution. Table S3 estimates the required sampling time needed to sample enough PM mass to reach the quantification limit for organics, sulfate, and nitrate. Assuming an ambient PM concentration of 10 µg m$^{-3}$ and a sampler flow rate of 2.5 L min$^{-1}$, less than 0.5 hours of sampling are required. This sampling resolution will significantly benefit the UAS sampling of PM.





**Data availability.**

All data sets including HR-AMS, ACSM, and SIMS mass spectral data are available upon request from Qi Zhang, dkwzhang@ucdavis.edu.

**Supplement.**

The document contains additional mass spectral datasets (for the ACSM and SIMS) and information on AMS data analysis with $^{34}SO_4$ as well as particle size distributions.

**Author contributions.**

CRN and QZ designed experiments and carried them out. FM and BS performed ambient sampling at PNNL and SGP. MAZ provided ACSM data. ZZ performed SIMS data analysis. CRN and QZ prepared the paper with contributions from all authors.

**Competing interests.**

The authors declare that they have no conflict of interest.

**Acknowledgements.**

This research was supported by, and data were obtained from the Atmospheric Radiation Measurement (ARM) User Facility, a U.S. Department of Energy (DOE) Office of Science User Facility managed by the Biological and Environmental Research Program and the DOE's Atmospheric System Research Program (Grant #DESC0022140). The SIMS analysis was performed on a project award (Award DOI: 10.46936/expl.proj.2021.60186/60008210) from the Environmental Molecular Sciences
Laboratory, a DOE Office of Science User Facility sponsored by the Biological and Environmental Research program under Contract No. DE-AC05-76RL01830. MAZ was supported by the Scientific Focus Area (SFA) Science Plan program that is supported by the Office of Biological and Environmental Research in the Department of Energy, Office of Science, through the United States Department of Energy Contract No. DE-SC0012704 to Brookhaven National Laboratory. CRN also acknowledges funding from the Jastro-Shields Research Award from the University of California at Davis.

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
