# Peer review of "Quantitative Chemical Assay of Nanogram-Level PM Using Aerosol Mass Spectrometry: Characterization of Particles Collected from Uncrewed Atmospheric Measurement Platforms"

_Atmospheric Measurement Techniques, 2022_

## Referee Comment (RC1)

*In this manuscript the authors report the development of a micronebulization-AMS (MN-AMS) technique that can provide quantitative analysis of nanogram level of organic and inorganic substances by utilizing an isotopically labeled internal standard ($^{34}SO_4^{2-}$). Its major advantage is the less requirement of liquid volume for stable aerosol generation. As a result, it will meet the needs of applying AMS on offline chemical analysis of weight-limited PM samples from uncrewed atmospheric measurement platforms (UxS). Overall the manuscript is well written and the analysis is fairly easy to follow. I really enjoyed reading this article because it provides enough details of the experimental design, and also because the authors take an effort to validate the methods via multiple comparisons to other techniques such as IC, ACSM and SIMS. Therefore, I strongly support the publication of this work. Below are several minor comments that I would like to further discuss with the authors.*

*Minor comments.*

*1) Lines 227-229: Comparisons are made between the standard atomizer and the micronebulizer by atomizing a solution of sucrose and ammonium sulfate. The mass spectra derived from each atomizer show a high degree of similarity. What are the concentrations of sucrose and ammonium sulfate applied in this solution? Did the authors try different inorganic-to-organic mass ratio for this validation? Since atmospheric aerosol particles usually exhibit different morphologies and have complex chemical compositions, a validation by using a more complicated solution extracted from atmospherically relevant particles may be worthwhile. If this kind of sample is not available, adding some SVOCs, nitrogenated organic compounds, or chlorine species into the current solution could be some options. This validation can be critical, since one focus of this study is the micronebulization system, and Sections 3.2 and 3.3 are field applications rather than standard lab validations.*

*2) Lines 299-308 in the main text and Figure S4 in the SI: I am wondering if it's reasonable to conclude "Overall, the chemical compositions of the filter extract and the impactor extract are similar". In Figure S4 c), the fractions of total organic and "Chl" look quite different between the filter and impactor samples. The contamination from methanol is mentioned as one reason for higher concentrations of organic matter in the filter sample. Could the authors help explain why filter extracts contain more methanol but have less total organic matter? Will more methanol residuals contribute to more total organic matter in Fig. S4 c)? The similarity is also evaluated in the Figure S4 b), and it looks like the normalized signals are more consistent to each other in the region of both x and y values < 0.015? Is this similarity mainly from the methanol signals? Is it possible to make a comparison without methanol-related signals? As the authors mentioned in lines 297-299, the comparison between the filter (PNNL_F8) and the impactor (PNNL_I3) assesses biases either in the sampling system or the extraction procedure. For me it's totally ok if two samples do not show apparent similarities due to many factors. Some random thinking. What about the RH of that day and the particle viscosities? Is it possible that the impactor preferably holds more viscous particles that have larger mass fraction of organic matter?*

*3) Lines 330-331: "The organic mass spectra patterns from the two approaches are similar too for the indicated sampling periods (Fig. 5 b-e and Figure S5 a-d)." Is there a way for us to quantify the similarity between mass spectra obtained from MN-AMS and ACSM? I did not see $r^2$ in Figure 5 b-e as noted in Figure S5. Also, is $r^2$ a good way to represent similarity? As in Figure S4 b), if we use $r^2$ to show similarity, what does the slope mean in the fitting? If we use the same method in Figure S4 b) to estimate the similarity between mass spectra of MN-AMS and ACSM, should we use a slope of 1 to do the fitting? If not, does the slope represent the systematic over- or under- estimation between those two methods?*

*4) Line 222: What temperature the spray chamber is heated up to? Will some dissociation reactions occur at this temperature?*

*5) Lines 236-238 in the main text and Figure S2 in the SI: I saw for the concentration of 1.75 mg $L^{-1}$ there is already a considerable fraction of particle mass outside the 100% transmission range of the AMS. I am wondering how large is this fraction and how this will influence the sensitivity of AMS? Will the transmission drop significantly once the particle diameter is below 100 nm? Also, what about the case for the concentration of 1 mg $L^{-1}$?*

*6) Lines 140-141: In Figure 1. b). Is it better to present different parts in number and list their names aside as a legend? Readers might not be able to clearly see the setup underneath the text of "Spray chamber heater". Is the condensation liquid reusable? This is hidden by the text on the graph.*

*Typos or Formats*

1) *Line 14 in the Abstract, "as low as 10 μL" is not consistent with line 380 "as low as 100 μL"*
2) *Line 193: The [X] item in equation (2) seems not quite right.*
3) *Line 228: "similarly" or "Similarity"?*
4) *In Figure 5(a), the pie charts can be rotated in a way so that the readers can make easy comparisons, especially for the green parts representing organics. Top boundary for the green portion can be vertical.*
5) *Figure S6 (b), "CH3NO2" should not be at m/z=59?*

---

## Author Comment (AC1)

Response to reviewer comments on "**Quantitative Chemical Assay of Nanogram-Level PM Using Aerosol Mass Spectrometry: Characterization of Particles Collected from Uncrewed Atmospheric Measurement Platforms**"

We thank the editor and the reviewers for their thoughtful and constructive comments and we have revised the manuscript accordingly. Listed below are our point-to-point responses (in **blue**) to the comments (repeated in black) and changes of the manuscript (in **red**).

**Responses to Reviewer #1:**

In this manuscript the authors report the development of a micronebulization-AMS (MN-AMS) technique that can provide quantitative analysis of nanogram level of organic and inorganic substances by utilizing an isotopically labeled internal standard ($^{34}SO_4^{2-}$). Its major advantage is the less requirement of liquid volume for stable aerosol generation. As a result, it will meet the needs of applying AMS on offline chemical analysis of weight-limited PM samples from uncrewed atmospheric measurement platforms (UxS). Overall the manuscript is well written and the analysis is fairly easy to follow. I really enjoyed reading this article because it provides enough details of the experimental design, and also because the authors take an effort to validate the methods via multiple comparisons to other techniques such as IC, ACSM and SIMS. Therefore, I strongly support the publication of this work. Below are several minor comments that I would like to further discuss with the authors.

Minor comments.

1) Lines 227-229: Comparisons are made between the standard atomizer and the micronebulizer by atomizing a solution of sucrose and ammonium sulfate. The mass spectra derived from each atomizer show a high degree of similarity. What are the concentrations of sucrose and ammonium sulfate applied in this solution?

The concentration of both sucrose and sulfate (from ammonium sulfate) are 1 mg L$^{-1}$. This information will be added to the text in section 3.1.1.

Did the authors try different inorganic-to-organic mass ratio for this validation? Since atmospheric aerosol particles usually exhibit different morphologies and have complex chemical compositions, a validation by using a more complicated solution extracted from atmospherically relevant particles may be worthwhile. If this kind of sample is not available, adding some SVOCs, nitrogenated organic compounds, or chlorine species into the current solution could be some options. This validation can be critical, since one focus of this study is the micronebulization system, and Sections 3.2 and 3.3 are field applications rather than standard lab validations.

We did not compare more complex solutions in both the standard atomizer and micronebulizer. Given the limited availability of filter and impactor samples, we were unable to analyze the PM extracts using both the standard atomizer and micronebulizer. While such an analysis could help with the validation of our technique, both the standard atomizer and micronebulizer use the same physical principles for aerosol generation and are unlikely to give significantly disparate results.

2) Lines 299-308 in the main text and Figure S4 in the SI: I am wondering if it's reasonable to conclude "Overall, the chemical compositions of the filter extract and the impactor extract are similar". In Figure S4 c), the fractions of total organic and "Chl" look quite different between the filter and impactor samples.

The lines in question are meant to refer only to the mass spectral similarity, not to the total PM composition. The text will be updated to clarify this point. There are more significant differences when looking at the fractional contribution of the different PM components, and the possible reasons for this are discussed in the text.

The contamination from methanol is mentioned as one reason for higher concentrations of organic matter in the filter sample. Could the authors help explain why filter extracts contain more methanol but have less total organic matter?

During filter extraction, more methanol was used compared to the impactor extractions, and the final concentration of methanol was higher in the filter extracts. While this is not explicit in the manuscript, the filter sample shown in Figure S4 had a higher absolute mass concentration of organic material. What is shown in Figure S4c is that the filter sample had, as you point out, a lower fractional contribution of organic material compared to the impactor sample.

Will more methanol residuals contribute to more total organic matter in Fig. S4 c)?

Yes. Preliminary work looking at the HR-AMS organic signal from different organic solvents at different concentrations did show a dependence on the final concentration of the solvent. Methanol concentrations ≤ 10 % show a low, but consistent, background organic signal. Above that concentration and the background signal begins to increase rapidly. Because of this preliminary data, all filter and impactor extracts had a final methanol concentration ≤ 10 % v/v.

The similarity is also evaluated in the Figure S4 b), and it looks like the normalized signals are more consistent to each other in the region of both x and y values < 0.015? Is this similarity mainly from the methanol signals?

I would agree that the correlation is tighter in this low-signal range ($< 0.015$). This region contains mostly higher m/z ions ($\sim> 80$ m/z) with a small number of low m/z ions. When examining the organic mass spectra of aqueous solutions containing 5 mg L$^{-1}$ sulfate and 10 % methanol, it is comprised almost exclusively of ions with m/z values $< 100$, and most of the signal is from lower m/z ions ($< 70$). Thus, while the methanol likely makes some contribution in the region you point out, the contribution is small. Rather, correlation in this region is more likely driven by similarities in the PM collected on both the filter and impactor.

Is it possible to make a comparison without methanol-related signals? As the authors mentioned in lines 297-299, the comparison between the filter (PNNL_F8) and the impactor (PNNL_I3) assesses biases either in the sampling system or the extraction procedure. For me it's totally ok if two samples do not show apparent similarities due to many factors.

Depending on the mass of PM collected (and thus the PM concentration in the extracted solution), this methanol subtraction can be difficult. The methanol mass spectrum at low concentrations (<10 %) is very consistent, but if the PM mass concentration is very low, the uncertainty with respect to subtraction could be quite large. This may have been the case with the PNNL samples as the collection times were short (3 hr). We have no data from other instruments

to compare the PNNL samples to, and very few samples for analysis, so we chose not to perform a methanol subtraction at this time. In the future, with more samples and more supporting measurements, this type of subtraction will be explored in more detail.

We agree that a lack of total agreement between different sample types is okay in this case. As you say, and as is discussed in the manuscript, there are a number of factors that will contribute to the differences in PM composition seen by the filter and impactor samples.

Some random thinking. What about the RH of that day and the particle viscosities? Is it possible that the impactor preferably holds more viscous particles that have larger mass fraction of organic matter?

This is an interesting question. We do not have data on RH at the time of sampling or particle viscosities for the PNNL samples in this manuscript, so we can only speculate.

Prior work comparing filter and impactor sampling does suggest a relatively strong RH-dependence on particle collection [1]. Particle bounce seems to be more of a problem for impactors compared to filters, and particle bounce becomes more significant at lower RH values. Given the unknown RH and particle viscosity at the time of sampling, it is possible the filter and impactor were sampling different particle populations (or, rather, the impactor was sampling a subset of the particles sampled by the filter).

3) Lines 330-331: "The organic mass spectra patterns from the two approaches are similar too for the indicated sampling periods (Fig. 5 b-e and Figure S5 a-d)." Is there a way for us to quantify the similarity between mass spectra obtained from MN-AMS and ACSM? I did not see $r^2$ in Figure 5 b-e as noted in Figure S5. Also, is $r^2$ a good way to represent similarity?

The $r^2$ referred to in the text was previously in the figure in question in earlier versions of the manuscript. After we decided not to include the $r^2$ values, this reference in the figure caption was accidentally left in. Thank you for pointing this out, it will be removed to avoid further confusion.

To your broader point about quantitative comparison between the MN-AMS and ACSM mass spectra: The quantitative comparison between the AMS and ACSM organic mass spectra, as based on a simple regression analysis, can be misleading due to the differences in sampling used between the real-time and offline analyses, as well as differences in mass spec fragmentation between the two instruments (noted in prior studies). Without the proper context of the differences in instrumentation, a low $r^2$ implies a worse correlation that may actually be the case. To avoid misinterpretation, we chose to remove the $r^2$ values from the figure and focus on the quantitative comparison in bulk composition as measured by the MN-AMS and ACSM.

Simple regression analyses, like the one referred to here that was ultimately removed from the manuscript, can suffer from biases (e.g. fitting can be heavily driven by high signal ions) partially eliminated by more complex statistical techniques. The spectral similarity index and contrast angle are more robust techniques, but they do not solve our original problem which was properly framing the correlation to avoid misinterpretations.

As in Figure S4 b), if we use r$^2$ to show similarity, what does the slope mean in the fitting? If we use the same method in Figure S4 b) to estimate the similarity between mass spectra of MN-AMS and ACSM, should we use a slope of 1 to do the fitting? If not, does the slope represent the systematic over- or under- estimation between those two methods?

In a simple regression analysis, the slope would give some indication of under/over-estimation. However, as discussed in the previous comment response, interpretation of both the r$^2$ and slope is not trivial in this case. Differences can arise from PM sampling, instrumental differences, and methods of data treatment. We certainly do not expect perfect agreement due to these differences and chose to remove the r$^2$ analysis entirely in favor of a qualitative comparison of the organic mass spectra derived from the MN-AMS and ACSM.

4) Line 222: What temperature the spray chamber is heated up to? Will some dissociation reactions occur at this temperature?

During PM extract sampling, the spray chamber was ~50 °C.

It is certainly possible some dissociation reactions, as well as evaporation, may occur for particularly temperature-sensitive/volatile species. For example, the comparatively high detection limit for nitrate in this study may be the result of nitrate evaporation occurring in the spray chamber. We are currently investigating the temperature dependence of $NO_3$ measurements in our system. For organic material, losses due to evaporation, as opposed to dissociation reactions, are a larger concern. Prior studies examining organic aerosol (OA) volatility show a wide range of results depending on OA source, aging, etc. Given the results shown in Figure 6, the 50 °C heating did not seem to introduce large measurement errors within our limited set of samples. This could be partially due to the short residence time of the particles within the cyclone. However, more work should be done to assess the spray chamber temperature dependence of different PM components as evaporation can be significant even at 50 °C.

5) Lines 236-238 in the main text and Figure S2 in the SI: I saw for the concentration of 1.75 mg L$^{-1}$ there is already a considerable fraction of particle mass outside the 100% transmission range of the AMS. I am wondering how large is this fraction and how this will influence the sensitivity of AMS? Will the transmission drop significantly once the particle diameter is below 100 nm? Also, what about the case for the concentration of 1 mg L$^{-1}$?

The 100% transmission range for spherical particles in the AMS inlet is ~70-500 nm. We have no measurements of particle shape derived from our micronebulizer, but we can still use this range as a guide. For the 1.75 mg L$^{-1}$ sample, the fraction of area outside of this range is less than 5% of the total area. Given this low fraction, the effect on the AMS sensitivity is likely to be correspondingly low.

Transmission efficiency of spherical particles begins to drop rapidly below ~70 nm, although there is still substantial transmission (>10 %) of particles from 30-70 nm. This does not begin to be a significant fraction of our generated particle distributions until the total PM liquid concentration drops below 1 mg L$^{-1}$. Below this concentration, the peak particle size drops below

100 nm and a significant fraction of generated particles is outside of the 100% transmission region. This would begin to significantly affect the method sensitivity.

6) Lines 140-141: In Figure 1. b). Is it better to present different parts in number and list their names aside as a legend? Readers might not be able to clearly see the setup underneath the text of "Spray chamber heater".

We thank the reviewer for the figure suggestion. While we tried not to occlude key aspects of the setup with the labels, this suggestion would improve overall readability. Below the "Spray chamber heater" text is a standard laboratory scissor jack.

Is the condensation liquid reusable? This is hidden by the text on the graph.

We have not attempted to reuse the condensation liquid. Excess liquid generated by the common, collision-based atomizers is reusable as this liquid volume is quite large. Here, even without spray chamber heating, the condensation volume is small relative to the already small sample volumes of < 1 mL. Condensation only became noticeable after continuously nebulizing multiple samples in a row. We could collect the condensation, but reusing it without dilution to increase the volume to a usable level (~100 uL with the setup as shown in Fig. 1b) is likely not feasible.

Typos or Formats

1) Line 14 in the Abstract, "as low as 10 µL" is not consistent with line 380 "as low as 100 µL"

The "100 µL" has been corrected to 10 µL.

2) Line 193: The [X] item in equation (2) seems not quite right.

It is indeed incorrect. The equation has been updated to the following:

$$[X]_{liquid} = [X]_{AMS} \times (\frac{[^{34}SO_4]}{[^{34}SO_4]_{AMS,adj}})$$

3) Line 228: "similarly" or "Similarity"?

The line should read "similarity". It has been updated as follows:

...high degree of similarity…

4) In Figure 5(a), the pie charts can be rotated in a way so that the readers can make easy comparisons, especially for the green parts representing organics. Top boundary for the green portion can be vertical.

We thank the reviewer for this suggestion. The current version has the alignment on the horizontal, starting with the red (sulfate) portion. Having the alignment be on the vertical, with the largest fraction of each sample being aligned (the green, organic portion), is a good suggestion. The figure has been updated so the fractions of each pie chart are aligned on the same vertical, starting with the green, organic portion.

5) Figure S6 (b), "CH3NO2" should not be at m/z=59?

Correct, this ion is actually C2H5NO. Thank you for this correction.

References

(1)     Nie, W.; Wang, T.; Gao, X.; Pathak, R. K.; Wang, X.; Gao, R.; Zhang, Q.; Yang, L.; Wang, W. Comparison among Filter-Based, Impactor-Based and Continuous Techniques for Measuring Atmospheric Fine Sulfate and Nitrate. *Atmos. Environ.* **2010**, *44* (35), 4396–4403.

---

## Author Comment (AC2)

Response to reviewer comments on "**Quantitative Chemical Assay of Nanogram-Level PM Using Aerosol Mass Spectrometry: Characterization of Particles Collected from Uncrewed Atmospheric Measurement Platforms**"

We thank the editor and the reviewers for their thoughtful and constructive comments and we have revised the manuscript accordingly. Listed below are our point-to-point responses (in **blue**) to the comments (repeated in black) and changes of the manuscript (in **red**).

**Responses to Reviewer #2:**

This manuscript presents a method to atomize small volumes of sample into an AMS for offline analysis. The spray is continuous and requires about 100 uL of liquid volume, and shows good comparison with online methods (ACSM). Offline analysis of aerosol samples is beneficial because it allows for characterizations to be made on samples that are significantly easier to collect (compared to flying an AMS). The paper is clear and well written and the work will be of interest to the readers of AMT. My main concerns are some needed clarifications and some corrections to statements made in comparison to prior work. Once these concerns are resolved, I recommend publication in AMT.

1) In the abstract and conclusions, the authors list a detection limit in nanograms. However, these samples are coming from solutions and it is not clear what the sample volumes are that these correspond to. If it is the same sample volume used everywhere, please make that more clear. Otherwise, please report the sample concentrations as well as the masses to improve reproducibility of the work.

Thank you for pointing out this lack of clarity. The sample volume is dependent on the syringe pump flow rate and HR-AMS averaging time. The HR-AMS averaging time was constant across all data points at 1 min. All UxS samples (i.e. all samples from PNNL and SGP) used the same flow rate of 50 µL min$^{-1}$, leading to a sample volume-per-HR-AMS data point of 50 µL. For the MN-AMS data from standard solutions (i.e. Figure 3), a variable flow rate was ranging from 13-53 µL min$^{-1}$. However, the data here is based on integrating the AMS-measured mass concentration for the entire 400 µL sample volume that was loaded into the syringe. So for the detection limits, the total volume-per data point on Fig. 3 is 400 µL although the sample concentration and nebulized volume-per-AMS run changes. This is described in lines 242-247. To improve the clarity regarding the detection limits, the following text will be added to the end of section 3.1.2:

Additionally, the low concentration samples analyzed here ([sucrose] = [SO$_4$] = 0.06 mg L$^{-1}$) sampled at the lowest usable flow rate (13 µL min$^{-1}$) were used to estimate the detection limits discussed later.

2) On page 3, it is noted that "Since the nebulization efficiency (i.e. the ratio between the mass detected by the AMS compared to the mass of solute nebulized) of the common aerosol generation systems is low, e.g., ~ 0.02% for an ultrasonic atomizer utilized by O'Brien et al. (O'Brien et al., 2019), liquid volumes of several milliliter and tens of micrograms of sample mass are usually required for continuous aerosol generation and AMS analysis (O'Brien et al.,

2019; Sun et al., 2011)." This statement is incorrect for O'Brien et al.. The efficiency is correct, however, the technique used a discrete injection, not continuous flow, and only 4-5 microliters of solution were used per injection. This should not be scaled to flows for a continuous injection as it misrepresents the method and over-estimates the volumes needed.

There is a lack of clarity to this sentence that we will correct. The nebulization efficiency does indeed refer directly to the data in O'Brien et. al. However, the "liquid volumes of several milliliter and tens of micrograms of sample mass" is meant to refer to the more commonly used collision-based atomizers (e.g. the TSI 3076). The statement is paraphrased from the introduction of O'Brien et. al, but is not meant to refer to their data specifically. You are correct that their low-volume, discrete injections should not be scaled to continuous flow systems with respect to sample volume/mass. The sentence you quote will be rewritten as follows to avoid this confusion:

The nebulization efficiency (i.e. the ratio between the mass detected by the AMS compared to the mass of solute nebulized) of aerosol generation systems is low, e.g., ~ 0.02% for an ultrasonic atomizer utilized by O'Brien et al. [1]. Additionally, liquid volumes of several milliliter and tens of micrograms of sample mass are usually required for continuous aerosol generation and AMS analysis [1,2].

3) The comparison of the UAS samples is welcome and interesting. In section 2.2, how were the blanks collected, handled, and prepared? In section 3.2 it is noted that the normalized blanks are subtracted from the samples. Were these mass subtractions only, or were the spectra subtracted as well? What did the blank spectra look like compared to the samples?

The blank filters and impactors were handled and prepared identically to the collected filters and impactors. A sentence mentioning this is currently in section 3.2, but will be moved to section 2.2 where the extraction of the filters and impactors is discussed in more detail.

For the blank subtraction, this was a mass subtraction only. The blank filters and impactors were similar to the mass spectra of solvent blanks comprised of $^{34}SO_4$ and methanol (at the same concentration range used for the filter/impactor extraction), suggested most of the background organic signal is derived from the methanol used during extraction and not from material adsorbed to the filters or impactors.

4) The use of isotopically labeled sulfate is a nice quantification method. Have the authors explored the ability to quantify with sulfate when ions like sodium or potassium are present in the sample? These can form salts with high vaporization temperatures and may be a concern for quantification.

We have not explored quantification with isotopically labeled sulfate when sodium or potassium are present in solution.

5) I appreciate the comparisons between he different HR spectra, but I would like more comparison with the online ACSM data. Figure S5 shows the ACSM data for I believe the same time periods as those in Figure 5. However, it is very difficult to directly compare. Please add a figure in the supplemental that is a direct comparison between the two (with the HR data unit

mass). The caption on Figure S5 also notes some r squared values that I cannot find in Figure 4. Please correct this.

The $r^2$ referred to in the text was previously in the figure in question in earlier versions of the manuscript but was later removed. The $r^2$ values have been added back to Figure S5. Additionally, as you suggest, we have improved the comparison between the AMS and ACSM organic mass spectra. Figure S5 now includes an overlay of the HR unit mass AMS data. The discussion regarding the comparison between the two instruments has been similarly expanded on. Section 3.3 now includes the following text:

The comparison between the MN-AMS and ACSM organic unit mass spectra is shown in Figure S5. The agreement between the MN-AMS and ACSM measurements is moderate ($0.5 < r^2 < 0.8$). However, it is important to remember the differences in PM sampling between the MN-AMS and ACSM (filter and impactor extraction of $PM_{2.5}$ vs real-time $PM_1$)and that the two instruments may have different sensitivities to certain organic species resulting in discrepancies for co-located AMS and ACSM measurements (e.g. [3]). Many of the most divergent ions measured in both instruments are $C_xH_y$ ions that have a significantly higher signal in the MN-AMS. This may suggest chemical differences in the $PM_{2.5}$ and $PM_1$ size regimes.

6) On page 15 no mention is made of differences that can be due to extraction and solubility of the samples. This may not be too large of a concern at SGP, but it may be a concern at other field sites and should be mentioned.

We agree. Issues with PM solubility were briefly discussed earlier in the manuscript when talking about the samples from PNNL, but it is worthwhile reiterating this point with the SGP samples, partially because these samples were discussed in much more depth than the PNNL samples, and because it certainly could account for some of the differences seen between the MN-AMS and ACSM datasets. The following text will be added to the manuscript in the section comparing the MN-AMS data to the ACSM data:

Last, it is possible that the extraction process, using both methanol and water, is a source of discrepancy between the MN-AMS and ACSM datasets as both organic and inorganic PM exhibits a range of solubilities in different solvents [4]. While the MN-AMS data resembled the online ACSM measurements to a high degree, differences in recovery of specific PM components when comparing offline to online results should be considered [5].

References

(1)   O'Brien, R. E.; Ridley, K. J.; Canagaratna, M. R.; Jayne, J. T.; Croteau, P. L.; Worsnop, D. R.; Hapsari Budisulistiorini, S.; Surratt, J. D.; Follett, C. L.; Repeta, D. J.; et al. Ultrasonic Nebulization for the Elemental Analysis of Microgram-Level Samples with Offline Aerosol Mass Spectrometry. *Atmos. Meas. Tech.* **2019**, *12* (3), 1659–1671.

(2)   Sun, Y.; Zhang, Q.; Zheng, M.; Ding, X.; Edgerton, E. S.; Wang, X. Characterization and Source Apportionment of Water-Soluble Organic Matter in Atmospheric Fine Particles (PM2.5) with High-Resolution Aerosol Mass Spectrometry and GC-MS. *Environ. Sci. Technol.* **2011**, *45* (11), 4854–4861.

(3)   Zhou, S.; Collier, S.; Xu, J.; Mei, F.; Wang, J.; Lee, Y.-N.; Sedlacek III, A. J.; Springston, S. R.; Sun, Y.;

Zhang, Q. Influences of Upwind Emission Sources and Atmospheric Processing on Aerosol Chemistry and Properties at a Rural Location in the Northeastern U.S. *J. Geophys. Res. Atmos.* **2016**, *121* (10), 6049–6065.

(4)     Mihara, T.; Mochida, M. Characterization of Solvent-Extractable Organics in Urban Aerosols Based on Mass Spectrum Analysis and Hygroscopic Growth Measurement. *Environ. Sci. Technol.* **2011**, *45* (21), 9168–9174.

(5)     Daellenbach, K. R.; Bozzetti, C.; Křepelová, A.; Canonaco, F.; Wolf, R.; Zotter, P.; Fermo, P.; Crippa, M.; Slowik, J. G.; Sosedova, Y.; et al. Characterization and Source Apportionment of Organic Aerosol Using Offline Aerosol Mass Spectrometry. *Atmos. Meas. Tech.* **2016**, *9* (1), 23–39.

---

## Author Comment (AC3)

Response to reviewer comments on "**Quantitative Chemical Assay of Nanogram-Level PM Using Aerosol Mass Spectrometry: Characterization of Particles Collected from Uncrewed Atmospheric Measurement Platforms**"

We thank the editor and the reviewers for their thoughtful and constructive comments and we have revised the manuscript accordingly. Listed below are our point-to-point responses (in **blue**) to the comments (repeated in black) and changes of the manuscript (in **red**).

**Response to Reviewer #3:**

This paper describes a micro-nebulization plus HR-AMS technique to analyze very small aerosol samples collected with the type of collectors used on unmanned aerial systems (UAS). This is a welcome addition to the use of the HR-AMS for offline analysis and makes in situ sampling with UAS systems a real possibility. The paper is fairly well written and should be accepted for publication after the authors address the following points.

Line 16 and Table S3: This data does not justify 3 significant digits, especially in the detection limit. I would use at most 2.

Thank you for this suggestion. We have reduced the significant figures to 2.

Line 25: I'm not sure what you mean by "with temporal and spatial resolution." The UAS sampling time in Table S1 is 15 hours over multiple days. I would delete this phrase.

This statement was meant to be taken as a more general comment, indicating that the opportunity for improving the temporal and spatial resolution for PM sampling can be improved with the MN-AMS technique. However, your comment makes it clear that this is not the message that is coming across. The text will be updated to the following to improve the clarity:

This study demonstrates the utility of combining MN-AMS with uncrewed measurement platforms to provide quantitative measurements of ambient PM composition.

Line 34-35 and throughout paper: Please format the citations properly, i.e., remove the extra parentheses and don't include the author's name if it is right before the citation.

We apologize for the incorrect citation formatting. The specific citations will be updated.

Lines 120-125: Since you include this data in the paper, please include the sampling dates and times in Table S1 and give the samples names. In Figure S7, you refer to SGP impactor 2, but it is not clear if you are referring to one sample or the average of multiple samples. Also, it is not clear if this is one stage (which size cut?) or multiple stages averaged together. Please specify in

the text that the four-stage impactor samples were collected at SGP. Delete "Note that" at the beginning of the last sentence.

The SIMS data presented in this manuscript was collected on 2021-11-16, 14:30-16:30. This overlaps with the end of sampling for the first impactor sample at SGP, not the second which is a typo in the caption for Fig. S7. This has now been corrected. The figure caption is referring to a single sample as we only had one sample (i.e. one filter or impactor) per sample listed in Table S1. The reference to a specific sample in the caption for Fig. S7 has been updated to "SGP_I1" to clarify this.

SIMS data from one stage of the impactor is presented in this manuscript as a supporting measurement. This was the $4^{th}$ stage with a size cutoff of 2.5 μm. Text indicating the SIMS sample was collected at SGP has been added to section 2.2.

Line 128: Please use one system of units.

The "5/32-inch" has been updated to "3.97-mm" for consistency.

Lines 129-132: Why is the filter extraction/sonication performed in two steps?

The first sonication is performed with only methanol to aid in the extraction of lower polarity organic material specifically. The second sonication uses a methanol/water (in the form of an aqueous solution of ammonium 34-sulfate) mixture to extract a wider (and more polar) range of material. There is prior work indicating that multiple extraction steps with different solvents can aid in the full extraction of PM adhered to filters (e.g. [1]). While we did not follow the methodology used in Bein and Wexler, 2015, this type of work was the impetus for performing two separate extraction steps. The limited number of samples (and total lack of replicate samples) precluded our ability to explore other extraction techniques.

Lines 144-148: In this description of the AMS, please mention that you were using N2 as the carrier gas for the nebulization. This is important for interpreting the mass spectra in Figure 2b.

We agree with the reviewer that this is necessary information for the interpretation of the mass spectra in Figure 2b. The text has been updated to note the use of $N_2$ as the carrier gas for nebulization.

Figure 2: Please use more different colors in (a) and (b) for SO4^2- and 34SO4^2-. e.g., red and black. It is hard to tell them apart.

Thank you for this suggestion as clarity of data presentation is extremely important. In this figure, $^{34}SO_4^{2-}$ will be changed to black for increased contrast.

Lines 162-167: What was the point of the SIMS analysis? Is it preferential to N containing organics? Are there any references for applying this technique to ambient aerosol samples? I do not understand what the SIMS analysis adds to this paper.

The SIMS data is meant to support the MN-AMS data. Specifically, we believe we are detecting N-containing organics in the MN-AMS data, which has some precedence in the literature (as noted in lines 353-354). In order to bolster the conclusion that these are real species we are detecting and not an artifact, as well as demonstrating consilience between the MN-AMS measurement and an independent measurement, we included the SIMS data for comparison.

SIMS analysis has been used extensively for analysis of ambient PM samples, as a recent review demonstrates [2]. Text and this citation will be added to section 2.3.4 indicating the prior use of SIMS for PM analysis. To the best of our knowledge, the SIMS analysis that was performed is not preferential to N-containing organics. This was the focus for comparison to the MN-AMS data only to reinforce our conclusion of N-containing organics at the SGP site. A broader comparison of AMS and SIMS is not applicable due to the large differences in ionization mechanisms between the two instruments.

Lines 176-184: This description in the text is not consistent with the modified frag table provided in the SI. For example, text says CO=CO2, but frag table has CO=0.75*CO2. Why the nonstandard multiplier? Text says that S is removed from the parameterization, but it is still in the frag table as calculated from SO and SO2. Please correct either the frag table or the text! I would also rephrase the end of the sentence on lines 181-183 which seems confusing for non-AMS experts. Instead of "parameterized to the 34SO2 and 34SO ions and the parameterizations for the S and 34S were removed" maybe this works better: "parameterized to the 34SO2 and 34SO ions. The signals for S and 34S were determined directly from the high-resolution fits. Direct measurement of S is possible when N2 is used as the carrier gas."

We apologize for the lack of consistency regarding the frag table in the SI. Regarding the CO parameterization, the text is correct but the figure is out-of-date. We did originally used the CO=0.75*CO2 parameterization based on prior work performed in our lab (which was originally referenced in the text but later removed). However, once we obtained the ACSM data, the AMS data was reanalyzed using the parameterization mentioned in the text as the helped to align the AMS and ACSM analyses. The Table S2 is an unfortunate carry-over from the earlier analysis, and will be updated to reflect the frag table used for the data presented in the manuscript.

Regarding the S and j33S parameterizations, both the text and the figure in the SI are accurate. The text may be unclear, however. It is only referring to the "HR_frag_sulphate_34" column in Table S2, not the frag table for ambient sulfate which was unchanged from the standard frag table parameterization for sulfate. The text will be updated to the following to clarify that the removal of S and j33S parameterizations refers only to the sulfate-34 data, and not to ambient sulfate:

This pattern was similar to the standard fragmentation pattern for sulfate except that the sulfate-associated $H_2O^+$ signal was parameterized to the $^{34}SO_2^+$ and $^{34}SO^+$ ions and the parameterizations for the S and $^{33}S$ signals were removed from the $^{34}SO_4$ fragmentation wave.

Line 193: Should that be [34SO4] instead of [X] in the numerator?

You are correct. We apologize for this error and the line has been updated as you indicate.

Lines 198-206: You describe the IC analysis of 34SO4 spiked samples, but you do not show any data for either the known laboratory solutions or the ambient samples. Please include a comparison of the IC and AMS SO4 for at least some samples. This could be a table in the SI and should be referred to in lines 225-226 where you mention additional validation with IC analysis.

IC analysis was performed mainly for preliminary validation for the use of $^{34}SO_4$ for quantifying $SO_4$ and for a more general assessment of any differences in behavior of $^{34}SO_4$ and $SO_4$, some of which is shown in Figure 2. A figure in the SI displaying the calculated $SO_4$ liquid concentrations determined for a series of standard solutions using both IC and MN-AMS will be included in the supplemental and descriptive text added to section 3.1.3 where the current Figure 2 is discussed.

Line 211 and elsewhere in the paper and SI: It's a "collison-type" atomizer, not "collision-type"

Thank you for pointing out this error. This is a rather interesting typo on our part. The use of the term "collision-type" was based on the term being used on the TSI webpage for the TSI 3076 atomizer which was used in this study (see Figure S1) as well as a number of prior studies (not directly relevant to this manuscript) using the term "collision-type". However, this may be a common typo across a number of sources and indeed the correct term does indeed appear to be "Collison-type". The text will be updated to fix this error.

Line 212 and line 216: I would remove editorializing comments like "sorely" and "apparently."

Thank you for the suggestion. This editorializing has been removed.

Line 222: What temperature is the spray chamber? Is it warm enough to evaporate NH4NO3 and lead to the lower reported recovery for NO3 in Table S3?

The spray chamber was ~50 °C. Based on prior work performed in our lab with a thermodenuder, this temperature would be sufficient to evaporate a significant proportion of ammonium nitrate in the aerosol [3]. While we have not explored temperature modification as a way to improve the $NO_3$ recovery, we believe the spray chamber temperature is a significant factor in the low $NO_3$ recovery.

Lines 243 to 247 and Figure 3: Why does the data roll over at higher mass loading? I would add a sentence describing how you calculate NE. It's also not clear how you get a single value for NE from curved data. Is it the slope? Average of the ratios, in which case you should include the standard deviation? Refer to Table S3 when you mention the NE values. I would reorganize the caption to Figure 3. By the time you get to referring to the ratio between the two values, it is not clear if you are referring to (a) or (b). And there's a typo – "as in Fig. 1b" should be "as in Fig. 3b."

It is not fully clear why the data rolls over at higher mass loadings. The highest mass loading data corresponds both to the most concentrated samples and also data collected using the highest syringe pump flow rate. We currently believe we are "maxing out" the nebulizer through a

combination of relatively high solute concentration (the rollover data points have ~10 mg L$^{-1}$ total solute concentration) and higher liquid flow rate. It is possible that higher solute concentrations are leading to larger particles which are more likely to impact on the walls of the spray chamber.

For the NE, the range given in the main text and the values in Table S3 are based on the "best case scenario" of high solute concentration. However, if we instead use the average of the ratios (which may better reflect variability that would be seen in ambient samples) the NE changes only slightly (~1.44 % ± 0.36).

Regarding the figure caption, we thank you for the typo correction and the suggestion for reorganizing the caption. Both comments have been taken into account in the corrected caption for Figure 3.

Lines 281-282: Figures 2a and 2b are reversed.

Thank you for the correction, the typo has been fixed.

Line 286: I would add "on the ground" to the heading to make it clear these were not UAS samples.

Thank you for the suggestion. The section heading has been amended as suggested.

Lines 294-5: Did you subtract total organic mass for the blanks or subtract the mass spectrum for the blanks? The latter might help resolve whether the methanol contaminants are the cause of the differences between the filter and impactor. It's a bit confusing that you had more organic in the impactor blanks, but less methanol.

The total organic mass for the blanks was subtracted, not the mass spectrum. The text has been modified to make this point clearer. We agree that the latter could help to resolve the issue with methanol contaminants, but the samples presented in this section likely had relatively low PM mass (given the short sampling duration), and attempts to perform a mass spectrum subtraction lead to highly uncertain data. This type of subtraction was performed for the samples collected at the SGP site which appeared to have a higher mass loading.

The impactor sample had a higher percentage of organic material, but lower total PM mass. Additionally, there are several differences in PM sampling between the filters and impactors collected at PNNL (noted in section 2.2) that could lead to further discrepancy in the collected PM.

Lines 311-317: I'm very confused by the UAS sampling. Table S1 has dates between 11/15 and 11/18, but Figure 4 shows a flight track on 11/13 and Figure 5 shows additional grey bars on 11/9 and 11/11. Is Table S1 incorrect? I'm also confused about whether you have one UAS filter sample or multiple filter samples because sometimes you use singular and sometimes plural in the text. Did you really fly a single filter over 9 days? How did you prevent adsorption of gas-phase species when the UAS was not flying? In Figure 5, is the pie comparison for the filter

using the ACSM data for only the indicated grey bar or for all the grey bars averaged together? Please clarify in the caption.

Table S1 contains an unfortunate error in the sampling times for the SGP_F1 sample. The sampling times effectively shown in Figure 5 are correct, and Table S1 has been corrected. Thank you for pointing out this confusing error. The flight track in Figure 4 does in fact correspond to an actual flight track used during sampling for the SGP_F1 sample.

The SGP_F1 filter sample was flown over 7 seven flight from 2021-11-08 to 2021-11-16 (not all days had flights, as reflected in Figure 5a). The filter was stored at -20 °C in between flights, but no further effort was made to prevent desorption of gas-phase species. This could be a further source of discrepancy between the MN-AMS and ACSM data not already discussed in the manuscript. While using the same filter in this manner is not an ideal situation, our purpose here was more of a proof-of-concept of the MN-AMS technique rather than an intensive comparison between the MN-AMS and ACSM data.

Thank you for pointing out the lack of clarity in the averaging of the ACSM "filter" data. It was indeed averaged over the indicated gray bars (i.e. when the UAS filter was being sampled). Section 3.3 has been modified to make this point clear:

Figure 5a shows the time series of ACSM-measured NR-PM$_1$ species, along with the corresponding sampling periods for the impactors and UAS filter samples during which the ACSM data was averaged.

Figure 4: I think you could move this figure to the SI. What is the blue blob around SGP? Is that UAS flight track or something else? And if it is UAS flight track, why is there a ring of much higher concentration around it? Is it a different altitude?

Thank you for the suggestion. We believe that Figure 4 provides useful context for the SGP samples, as the location is discussed in section 3.3 and is relevant to some interpretation of the data presented in both Figure 5 and Figures S5-7. For these reasons, Figure 4 was included in the main text. However, we will consider moving it to the SI given that the flight track is only relevant to a single flight for the SGP_F1 sample and particle concentration data shown is not discussed.

The blue blob around the SPG site is displaying the flight track. This is admittedly not very clear given the tight, nearly overlapping passes performed over SGP. More details about the flight tracks can be found in [4], as cited in section 2.2.

Figure 5: It would be easier to compare the pies if you start organics at the top. You could also size the pies by the mass loading. In the caption, delete one of the uses of "offline" in the first sentence.

Thank you for the suggestion about lining the organics on top. We agree this helps to improve the clarity and the figure has been modified as such. We looked at sizing the pie charts by PM loading, but given the amount of data already presented in Figure 5, we believe that adding

additional information would make the figure too busy. Also, this information, while helpful to see, would be redundant in Figure 5 as it is explicitly presented in Figure 6.

Line 353: Somewhere earlier you should mention that the Q-ACSM is a PM1 and a unit mass resolution instrument.

This is indeed an important detail to note. The fact that it is a PM1 instrument is mentioned at the first mention of the ACSM in lines 323-325. That is provides unit mass data, however, was not stated. This detail has been added to lines 323-325.

Line 381: "a small number of UAS-collected" suggests you analyzed multiple filters, but I think you only analyzed one. Please correct this.

You are correct that only one UAS-collected sample (SGP_F1) was analyzed in this study. The text in question was meant to refer to the use of the same sampling equipment used aboard the UAS and for the ground samples. However, this is not clear as written. The text has been updated to the following:

As a proof-of-concept, a small number of PM samples were collected using UAS sampling instrumentation and one sample was collected aboard a UAS and were analyzed using the MN-AMS technique.

Figure S2 a) caption: The description of the change in size distribution is not correct and not consistent with the text in lines 236-8 where you say that decreasing the concentration below 1 mg/L causes the size distribution to become too small to be effectively transmitted to the AMS lens. Please correct the caption.

The lines of the figure caption in question were referring to the mode diameters in Figure S2a. This was not stated, though, leading to a discrepancy between the figure caption and lines 236-238 as you point out. The relevant line in the figure caption has been changed to the following to avoid this confusion:

…The particle size distribution shifts to lower diameters as the total solute concentration decreases…

Figure S3: This figure is very confusing. What is the data in between organics and SO4? It is not identified in the legend. Please identify the lines in the legend!  It would help to scale the zeros to the same place on the left and right axes.  I'm going to assume that the unidentified trace is AMS SO4, in which case it looks like the ratio of the AMS Org/SO4 is 20:1, even though the solution concentrations are 5:1. That would suggest much lower NE for SO4 than Org, which is not consistent with Figure 3a. I don't understand the caption to Figure S3 – how is this a range of solute concentrations? It looks like just one. Presumably, this data corresponds to a single point in Figure 3. Which one? There's a significant variation in the AMS Org signal (from 0 to 50 ug/m3) across this time series. Do you have an explanation? When analyzing ambient samples, do you integrate across the entire nebulization? Or do you use the region where the AMS signal is stable?  For the left axis, how are you measuring the solution concentration and why does it decrease at the start of the nebulization? You also need to correct the description of this Figure in

the text (lines 258-261) which refers to comparing both AMS modes and not what is actually in Figure S3 (sample data for one point in Figure 3).

The data in between the organic and SO4 solution concentrations is the HR-AMS-measured concentration of both $SO_4$ and $^{34}SO_4$. This is currently labeled in the figure legend. We believe the confusion is coming from both the similar colors used for $SO_4$ and $^{34}SO_4$ and the fact that the data is fully overlapping on the x-axis and nearly overlapping on the y-axis (making the two datasets difficult to visualize). Lines are used as the markers for both, but the $^{34}SO_4$ marker was a diagonal line in order to make it distinguishable from the $SO_4$ data. To be consistent with a previous comment and to improve the clarity, the $^{34}SO_4$ has been changed to black and vertical lines, with different sizes used to make it more distinguishable from the $SO_4$ data.

Regarding the potential discrepancy with NE values with organics and $SO_4$, the confusion is the result of an unfortunate error on our part. The data presented in Fig. S3 is part of a much longer wave of data from experiments involving different ratios of organics, $SO_4$, and $^{34}SO_4$. When examining a different segment of the data that had a different ratio of $SO_4$ to $^{34}SO_4$, an offset to the y-axis was applied (as a quick check to see the alignment of the HR-AMS measured mass concentration of $SO_4$ and $^{34}SO_4$). The offset was mistakenly left on, leading to a visually lower (and as you point out, much lower than it should be) NE compared to organics and what is presented in Figure 3. The offset was purely a visualization effect, which is why the $^{34}SO_4$ normalized concentration also presented in Fig. S3 still gives the expected 5:1 ratio of organics-to-$SO_4$. We apologize for this oversight. The figure was been corrected, and the actual ratio of organics-to-$SO_4$ is ~4.2, still lower than expected but reasonable considering the additional uncertainty associated with the open – closed calculation needed to determine the diff signal for Fast MS mode.

Regarding the figure caption, it is unfortunately confusing as written. The data in Figure S3 is from a single solution, however the figure caption was making a broader statement (discussed in the main text) about the AMS-signal stability while using the micronebulization system and Fast-MS mode sampling. The figure caption will be modified to the following to avoid this confusion:

The HR-AMS-measured mass concentration of different component is highly reproducible using very low sample volumes (~53 µL) using the Fast-MS mode.

The data presented in Figure S3 is in no way connected to the data in Figure 3. They were produced using solutions of different composition (with no overlap) and using different sampling modes on the AMS (Fast-MS for Figure S3 and Gen-Alt in Figure 3).

The variation at the beginning and end of the sampling period shown in Figure S3 is likely from the start and end of liquid flow into the nebulizer. In our preliminary work, we consistently noticed spikes in the AMS signal when the liquid flow was stopping (either from the syringe pump being turned off or the solution running out). The rise is signal seen at the start of Figure S3 only shows the beginning of the liquid entering the nebulizer starting to form the particles measured by the AMS, which can be seen much more clearly when using Fast-MS mode and very short averaging times. However, in between these two states (liquid starting and stopping entering the nebulizer), the AMS-measured signal is very stable when using Fast-MS mode. This

is not stated in the main text as a more detailed analysis of the Fast-MS mode data was not a focus of our work at the time, and they key point for us was the stability and accuracy of the $^{34}SO_4$-normalized signal for both organics and $SO_4$.

For analysis of the ambient samples, only regions of AMS data with stable data were used. All ambient samples presented here were sampled using Gen-Alt mode with 1 min averaging. In this mode, given the small extraction volumes used for the ambient samples, we do see a similar rise and fall of the AMS signal when the solution is starting to enter the nebulizer and running out (although given the longer averaging time compared to the Fast-MS mode, the effect is not as dramatic). Only data points in between the rising and falling signal (i.e. where the signal was stable) were used.

The left axis is the $^{34}SO_4$-normalized signal, determined as described in section 2.4. For clarity, the axis label will be updated to "$^{34}SO_4$-normalized solution concentration (mg $L^{-1}$)" to be more consistent with the nomenclature used elsewhere in the manuscript.

Thank you for pointing out the inconsistency in lines 258-261. These lines were meant to make the broader point that the AMS-measured signal is stable in both modes (which is true despite such data for the Gen-Alt mode not being explicitly presented in the same manner as the Fast-MS mode. The small standard deviations shown in Figure 3a support this claim.) The text will be updated to the following:

As shown in Figure S3, for MN-AMS setup reported here, the Fast-MS mode provides highly reproducible measurements of different chemical components in the liquid sample and the liquid concentration of organics and sulfate measured using the Fast-MS mode are accurate when normalized to the known concentration of $^{34}SO_4$.

Figure S4: Use the same units on the y-axis of (a) and the axes of (b). One is in percent and the other is fraction. Please include the mass loadings for the species as well as the fractional contribution in (c). Please note in the caption that this comparison is for samples PNNL_F8 and PNNL_I3. You have other pairs that are very similar in day/time, e.g., PNNL_F7 and SGP _I2. What does that comparison look like? You could also compare the sum of PNNL_F1 through F6 with PNNL_I1. What does that look like? I think this is worth a comment in the text.

Regarding the units of Figure S4 a,b, the units are the same. The axis labels on b will be updated to note that they are a percentage of the total organic signal.

More detailed, quantitative analysis was reserved for the SGP samples specifically as we had supporting, independent measurements that comparisons could be made to (e.g. ACSM, SIMS data). Including mass loadings in Figure S4 may serve to add confusion to the purpose of the data in Figure S4, which is meant to be a qualitative comparison between the filter and impactor sampling at PNNL showing how the differences in PM sampling and extraction techniques may lead to differences in measured organics and inorganics.

It is true that the PNNL_F7 and PNNL_I2 (not SGP_I2) were sampled during similar time periods. Only data from PNNL_F8 and PNNL_I3 were shown in Figure S4 as the sampling

periods were purposefully fully overlapped, giving us the best look at the effects of sampling and extraction techniques on the observed chemistry. However, the comparison for the PNNL_F7 and PNNL_I2 samples were about as similar as the PNNL_F8 and PNNL_I3 samples. This comparison was excluded to avoid redundancy and unnecessarily lengthening the manuscript.

We did not attempt to sum PNNL_F1 through F6 and compare that to PNNL_I1, although such a comparison could be warranted given the closely overlapping sampling periods. Again, the most reasonable comparison is between PNNL_F8 and PNNL_I3, which were sampled for precisely the same time period and are a single filter being compared to a single impactor, rather than being sampled at (slightly) different time periods or being the sum of multiple filters compared to a single impactor. We did not feel that a more exhaustive analysis of the PNNL samples was warranted, given the greater abundance of independent measurements available at the SGP site.

Figure S5: It is odd that the Q-ACSM MS for Impactor 3 shows a lot more signal at higher m/z's, but this is not reflected in the HR-AMS data. Or maybe this is an artifact of the m/z transmission efficiency calibration in the Q-ACSM data? Did something change in the way the Q-ACSM was operating?

We are unaware of any changes in the operation of the Q-ACSM during the entire sampling period, although we agree this discrepancy is odd. Unfortunately, due to the low PM loadings observed during the sampling period for SGP_I3 (see Figure 5a), a number of factors may be at play here. The difference in ambient particle sizes sampled directly by the ACSM and collected onto the impactor may be a factor and more noticeable at very low PM mass loadings. Additionally, due to the low PM mass loading, the subtraction of the impactor blank is more significant for this impactor sample compared to the other SGP samples, leading to more uncertainty in the subtracted data.

Figure S6: It looks like the CxHyNO peak at m/z 59 is mislabeled as CH3NO2. It is called C2H5NO in the text (line 357) and in Figure S7b. I am also concerned by the fit to m/z 59 that you show in Figure S7b. Did you really not fit the C3H7O and C2H3O2 ions? You can't just arbitrarily leave out the organic ions because you want to see CxHyNO. Also, C2H7N2 is a very strange ion and not in the CxHyN series that you are observing. I suspect that is really C3H7O. Please fit m/z 59 with the correct set of possible ions and then redo the MS in Figure S6 and Figure 5 (b-e).

Regarding the text reference in line 357, you are correct that this is a typo in the figure as C2H5NO is the correct label. It has been corrected in the figure.

Regarding the fitting of m/z 59, there are two issues here. Both ions you suggest (C3H7O and C2H3O2) were in fact fit, but are not displayed in Figure S7b. They are not displayed for what was meant to be clarity as the signal intensity for both ions is extremely low (when C2H7N2 is also fit) and not obvious at the scale presented in Figure S7b. However, this does indeed imply we did not fit them so they will be added back to Figure S7b. We agree that one cannot arbitrarily include or exclude ions to obtain a wanted result. While the C2H7N2 ion is strange, it was included as the SIMS data suggested the presence of both C3H7O and C2H7N2. Additionally, there are other CxHyN2 ions present (as suggested by the SIMS data), but at

notably lower AMS signal intensities making them not clear at the scales presented in Figure 5 or Figure S6.

Figure S7: Which AMS data are you using for the comparison with the SIMS data? Weren't they collected at different times? Does the SIMS preferentially detect CxHyN, but not CxHyNO? I don't understand how you have scaled the axes for the two signals. If your point is that the SIMS signal is comparable to the AMS CHN signal, then it seems like you should use the same relative scaling between right and left axes on all panels.

The AMS data used in Figure S7 is from SGP_I1 (originally mislabeled as SGP_I2). The SIMS sample shown here was collected on 2021-11-16 from 14:30 to 16:30 UTC. This has it overlapped almost entirely with SGP_I1, although SGP_I1 was sampled for a much longer period.

The scaling for the SIMS and AMS data has no purposeful connection as presented in Figure S7. Both datasets are presented as the raw signal intensities. We do not mean to imply any connection between signal intensities, rather we are trying to demonstrate that we can observe similar ions using these two independent measurements (with the focus being more on nitrogen-containing organics). Given the lack of similarities between on the SIMS and AMS signals are determined, we chose not to modify the scaling of either and present them simply as their raw signal intensities.

References:

(1)     Bein, K. J.; Wexler, A. S. Compositional Variance in Extracted Particulate Matter Using Different Filter Extraction Techniques. *Atmos. Environ.* **2015**, *107*, 24–34.

(2)     Huang, D.; Hua, X.; Xiu, G.-L.; Zheng, Y.-J.; Yu, X.-Y.; Long, Y.-T. Secondary Ion Mass Spectrometry: The Application in the Analysis of Atmospheric Particulate Matter. *Anal. Chim. Acta* **2017**, *989*, 1–14.

(3)     Yu, L.; Smith, J.; Laskin, A.; Anastasio, C.; Laskin, J.; Zhang, Q. Chemical Characterization of SOA Formed from Aqueous-Phase Reactions of Phenols with the Triplet Excited State of Carbonyl and Hydroxyl Radical. *Atmos. Chem. Phys.* **2014**, *14* (24), 13801–13816.

(4)     Mei, F.; Pekour, M.; Dexheimer, D.; de Boer, G.; Cook, R.; Tomlinson, J.; Schmid, B.; Goldberger, L.; Newsom, R.; Fast, J. Observational Data from Uncrewed Systems over Southern Great Plains. *Earth Syst. Sci. Data Discuss.* **2022**, *2022*, 1–25.